# Multimodal Negative Learning

**Baoquan Gong**[*] **Xiyuan Gao**[*] **Pengfei Zhu** **Qinghua Hu** **Bing Cao**[†]

School of Artificial Intelligence, Tianjin University, Tianjin, China

Haihe Lab of ITAI

{gongbaoquan, gaoxiyuan, zhupengfei, huqinghua, caobing}@tju.edu.cn

## Abstract

Multimodal learning systems often encounter challenges related to modality imbalance, where a dominant modality may overshadow others, thereby hindering the learning of weak modalities. Conventional approaches often force weak modalities to align with dominant ones in *"Learning to be (the same)" (Positive Learning)*, which risks suppressing the unique information inherent in the weak modalities. To address this challenge, we offer a new learning paradigm: *"Learning **Not** to be" (Negative Learning)*. Instead of enhancing weak modalities' target-class predictions, the dominant modalities dynamically guide the weak modality to suppress non-target classes. This stabilizes the decision space and preserves modality-specific information, allowing weak modalities to preserve unique information without being over-aligned. We proceed to reveal the multimodal learning from a robustness perspective and theoretically derive the *Multimodal Negative Learning* (**MNL**) framework, which introduces a dynamic guidance mechanism tailored for negative learning. Our method provably tightens the robustness lower bound of multimodal learning by increasing the Unimodal Confidence Margin (UCoM) and reduces the empirical error of weak modalities, particularly under noisy and imbalanced scenarios. Extensive experiments across multiple benchmarks demonstrate the effectiveness and generalizability of our approach against the competing methods. The code is available at `https://github.com/BaoquanGong/Multimodal-Negative-Learning.git`.

## 1 Introduction

Multimodal learning has become a cornerstone in many real-world applications, such as autonomous perception [1], medical diagnosis [2], and human-computer interaction [3]. By integrating information from multiple sources, such as vision, audio, and text, multimodal systems aim to improve performance and generalization. However, multimodal data often exhibit a significant imbalance between modalities due to noise, lack of information, or sensor heterogeneity [4]. Unimodal prediction accuracy is widely used to detect modality imbalance, which is a simple and effective metric [5, 6, 7], but it is inherently sensitive to noise and perturbations [8]. This vulnerability often leads to fragile performance in practice, limiting the reliability of such definitions in real-world deployments.

Extensive prior studies, especially those based on late fusion, also known as decision-level fusion strategies, have attempted to alleviate modality imbalance. Common approaches include enhancing the predictive performance of weak modalities through aggregating independently trained modality-specific classifiers [9, 10], confidence-based weighting [5], adaptive ensembling [11], or knowledge distillation [12, 13] from dominant modalities. While effective in certain scenarios, these methods often implicitly aim to align weak modalities with dominant ones in terms of prediction accuracy. This

---

[*]Equal contribution.

[†]Corresponding author.

39th Conference on Neural Information Processing Systems (NeurIPS 2025).

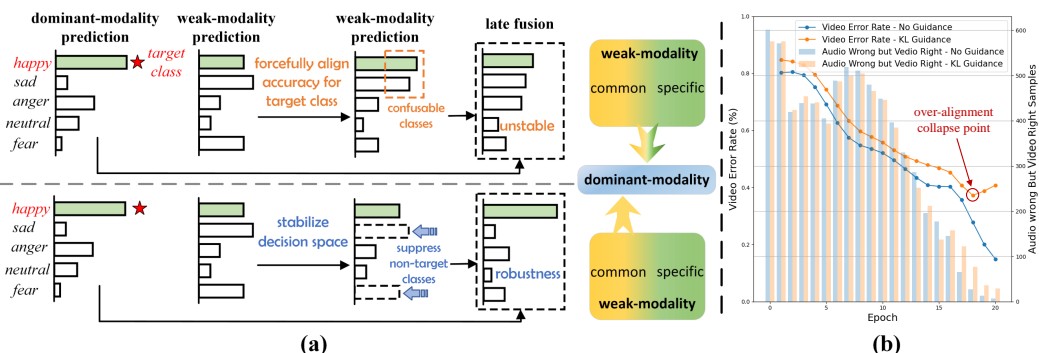

Figure 1: **(a)** Illustration of conventional (top) vs. our (bottom) strategies: instead of improving target prediction of weak modalities, we stabilize their decision space by suppressing non-target classes, which can preserve the modality-specific information and enhance robustness. **(b)** Empirical evidence: forced alignment can degrade weak modality (*Video*) predictions on certain samples, highlighting the risk of losing modality diversity. Where *KL Guidance* stands for the forced alignment method.

over-alignment may lead to several issues: (i) suppressing the distinct and complementary information encoded in weak modalities; (ii) risking error propagation, as dominant modality errors are amplified when weak modalities follow them blindly [14, 15]. More importantly, we conduct a statistical analysis illustrated in Fig. 1 (b), showing that traditional alignment-based strategies may even harm weak modalities. Specifically, we observe that samples originally predicted correctly by weak modalities (e.g., *Video*) but incorrectly by dominant modalities (e.g., *Audio*) become misclassified after being guided by fixed unidirectional KL guidance, as training progresses and eventually leads to what we term an *over-alignment collapse point*, where weak modalities lose their original predictive advantages due to excessive conformity [16].

In this paper, we propose a novel perspective: rather than requiring weak modalities to select the correct class, we teach them to eliminate implausible ones, forming a negative learning process. This paradigm shift is motivated by the intuition that it is often easier to rule out wrong answers than to select the right one [17, 18], especially when data quality is limited. As shown in Fig. 1 (a), unlike conventional methods that force weak modalities to identify the target class, they are often highly sensitive to noise and perturbations. Our approach enhances robustness by stabilizing the decision space, making the fusion process more resilient to uncertainty. Decision space instability refers to a model's high sensitivity to input perturbations near the decision boundary, where even minor changes may cause large shifts in prediction. By mitigating this instability, our method not only improves robustness but also preserves modality-specific information.

By allowing dominant modalities to guide weak ones through negative learning, we gain two major advantages: (1) we stabilize the decision space, thereby improving the robustness and consistency of the final prediction, making it less sensitive to noise and better at resisting perturbations; (2) we reduce the performance gap between modalities, effectively mitigating modality imbalance. This elimination-based view of weak modalities transforms them from noisy distractions into active agents of uncertainty reduction, which is especially useful in safety-critical or imbalanced scenarios.

To support this view, we introduce a *learning not to be* strategy for multimodal learning that shifts the focus from only improving weak modality predictions to reducing uncertainty over non-target classes. We establish a theoretical guarantee on the robustness lower bound of decision-level fusion, showing that this uncertainty suppression leads to more reliable performance. Furthermore, we demonstrate that the empirical error of weak modalities can be significantly reduced under this strategy, especially in noisy or imbalanced scenarios. In other words, our method not only enhances multimodal cooperation robustness under perturbations, but also narrows the performance gap between modalities. We summarize our main contributions as follows:

- We provide an intuitive and rigorous multimodal learning paradigm from the perspective of robustness. Under the theoretical analysis, we propose a new negative learning paradigm, stabilizing the decision space by instructing non-target classes of the weak modalities to learn from the dominant one.

- Building on our theoretical finding, we derive a new Multimodal Negative Learning (MNL) framework based on the Unimodal Confidence Margin (UCoM). This offers theoretical guarantees to tighten the robustness lower bound from multimodal learning, effectively mitigating modality imbalance and boosting robustness.

- Our model is flexible and compatible with a wide range of late fusion methods without introducing additional inference overhead. Extensive experiments confirm its practical effectiveness and generalizability in challenging multimodal scenarios.

## 2 Related Work

### 2.1 Imbalanced Multimodal Learning

Modality imbalance is a common challenge in multimodal learning, where different modalities vary in quality, completeness, and reliability. This issue is present across all fusion strategies: early fusion [19, 20], intermediate fusion [6, 21, 22], and late fusion [23, 24]. Among these, late fusion remains one of the most widely adopted paradigms due to its modular design, interpretability, and strong compatibility with unimodal pre-trained models [25]. Due to the lack of feature-level compensation or cross-modal interaction, late fusion tends to amplify the influence of strong modalities while weakening the role of lower-quality ones. Consequently, predictions from disadvantaged modalities often carry little weight or even introduce noise into the final decision [26]. To address this, a variety of methods have been proposed to alleviate imbalance in late fusion settings. These include confidence-based weighting [27, 28], modality dropout or gating [29], and adaptive ensembling strategies [11], which aim to dynamically suppress or correct low-quality predictions. However, the prevailing philosophy in these works is to improve the predictive accuracy of weak modalities for the target class, attempting to bring them closer to their stronger counterparts, which can easily cause modality-specific information loss. Inspired by the intuition that "*Learning not to be*: ruling out wrong answers is often easier than identifying the correct one [17, 18]", we propose a novel perspective: leveraging dominant modalities to assist weak modalities in identifying and suppressing non-target classes. By stabilizing the decision space of weak modalities in this way, we reduce their exposure to noise and uncertainty, and enhance their utility in the final decision. This approach not only promotes alignment between weak and dominant modalities, but also preserves the distinctive information of weak modalities, enabling more diverse and robust decision fusion.

### 2.2 Robustness in Multimodal Cooperation

Robustness has long been a critical topic in multimodal learning due to the inherent sensitivity of cooperation to perturbations such as data noise, label noise, and incomplete modality [30, 31]. These vulnerabilities stem from the heterogeneity of the modalities and their varying reliability under real-world scenarios. Existing studies address this by incorporating uncertainty modeling [21], designing robustness-aware fusion strategies [5, 6]. Some works also attempt to quantify multimodal robustness through novel evaluation metrics [16], or connect it with generalization error bounds [32, 33]. Collectively, these studies have highlighted a critical insight: the robustness of a multimodal system can be bottlenecked by a single weak modality [34], revealing a strong link between modality imbalance and overall system reliability. However, most of these approaches emphasize prediction-consistency modeling, often overlooking the role of decision space instability—especially the impact of non-target categories. This leads to more complex architectures with limited generalizability across tasks with different class granularity [35]. In contrast, our method improves robustness from a decision perspective: by identifying and suppressing non-target categories, we stabilize the effective decision space and enhance resistance to noisy or conflicting predictions. This contributes to a tighter theoretical robustness lower bound for decision-level fusion, providing both practical generalization and theoretical justification.

## 3 Method

In this section, we first clarify the basic setup of the multimodal late fusion system and extend the multimodal robustness lower bound to the late fusion framework. We then introduce Multimodal Negative Learning (MNL), which reduces uncertainty in non-target classes and tightens the robustness

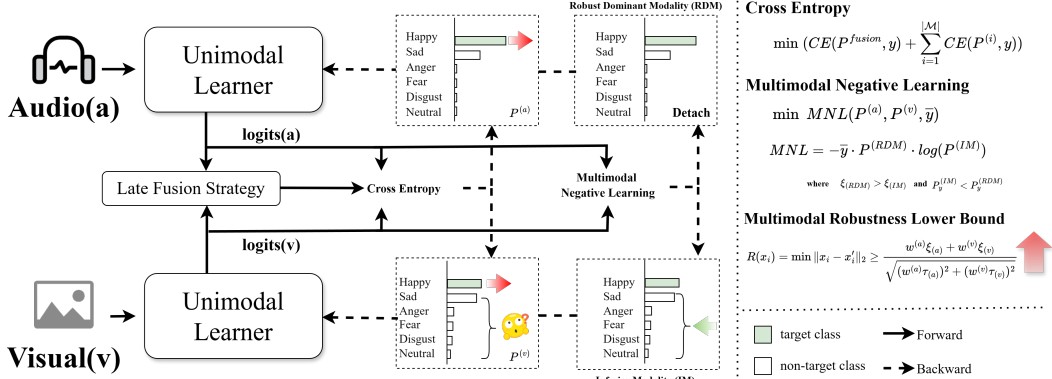

Figure 2: After cross entropy optimization, each unimodal learner updates its parameters based on ground-truth supervision, overlooking non-target class optimization, especially problematic for inferior modalities, which introduce noise during fusion. With the MNL, the robust dominant modality guides the inferior one by reducing uncertainty over non-target classes, thereby enhancing multimodal robustness with a larger UCoM.

lower bound in multimodal learning by enhancing the UCoM. Finally, we present the multimodal training strategy that incorporates the MNL.

## 3.1 Basic Setting

Given multimodal tasks, we denote the set of modalities by $\mathcal{M}$, where $|\mathcal{M}|$ represents the cardinality of $\mathcal{M}$. Without loss of generality, the training data points are denoted as $\mathcal{D}_{train} = \{(x_i, y_i)\}_{i=1}^{N} \subset \mathcal{X} \times \mathcal{Y}$, where $N$ is the sample size of $\mathcal{D}_{train}$, $x_i = \{x_i^{(1)}, \ldots, x_i^{(|\mathcal{M}|)}\}$ represents the input for the $i$-th sample across all modalities, and $y_i \in \mathcal{Y}$ denotes the corresponding label. In the late fusion framework, we define the logit output of the unimodal model is given by $f^{(m)}$. Depending on the specific fusion strategy employed in late fusion, the unimodal logits are either averaged or dynamically fused. The fusion result is represented as $f(x)$, which is formally defined as:

$$f(x) = \sum_{m=1}^{|\mathcal{M}|} w^{(m)} f^{(m)}, \tag{1}$$

where $w^{(m)} \geq 0$ and $\sum_{m=1}^{|\mathcal{M}|} w^{(m)} = 1$. Static late fusion assigns equal weights to all modalities, whereas dynamic late fusion allows the weights to vary across samples. Overall, late fusion is favored for its flexibility, robustness, and interpretability [36].

## 3.2 Robust Lower Bound for Multimodal Learning with Late Fusion

Inspired by [16] and following their definition of multimodal robustness, we extend the lower bound of multimodal robustness to the case of late fusion. Specifically, when the multimodal model $f$ correctly classifies the sample $x_i$, we introduce the multimodal robustness radius for the sample $x_i$:

$$R(x_i) = \min \|x_i - x_i'\|_2 \quad \text{s.t.} \exists j \neq y, \ f(x_i')_y = f(x_i')_j \tag{2}$$

where $\forall k \neq y, \ f(x_i)_y > f(x_i)_k$, $f(x_i)_k$ denotes the logit of class $k$ and $x_i'$ denotes the adversarially perturbed sample. Given the ground-truth label $y_i$ and its nearest competing class $j$, this defines the smallest perturbation, denoted as $x_i - x_i'$. Any perturbation smaller than this, i.e., within the multimodal robustness radius, can thus be reliably defended. For simplicity, for a unimodal logit output $f^{(m)}(x_i^{(m)})$, we introduce the Unimodal Confidence Margin (UCoM) as follows:

$$\xi_{(m)}^j = f^{(m)}(x_i^{(m)})_y - f^{(m)}(x_i^{(m)})_j \tag{3}$$

where $y$ is the target class and $j$ is the most probable class among the non-target classes. Unless otherwise specified, we omit the explicit subscript of the competing class $j$ for simplicity, and denote it compactly as $\xi_{(m)}$. We analyze the UCoM. A larger margin indicates that the unimodal modality is more reliable in distinguishing between these two classes.

For the competing class $j$ in the final fused result, if there exists a perturbed sample $x_i'$ of $x_i$ such that $f(x_i')_y = f(x_i')_j$ (i.e., a critical state), then for the original sample $x_i$, the multimodal CoM between class $y$ and the competing class $j$ is given by:

$$f(x_i)_y - f(x_i)_j = \sum_{m=1}^{|\mathcal{M}|} w^{(m)} \cdot \xi_{(m)} - \sum_{m=1}^{|\mathcal{M}|} w^{(m)} \cdot \xi'_{(m)} = \sum_{m=1}^{|\mathcal{M}|} w^{(m)} \cdot (\xi_{(m)} - \xi'_{(m)}) \quad (4)$$

where $\sum_{m=1}^{|\mathcal{M}|} w^{(m)} = 1$ and $\sum_{m=1}^{|\mathcal{M}|} w^{(m)} \cdot \xi'_{(m)} = 0$. Furthermore, we introduce the Lipschitz constant $\tau_{(m)}$ [16, 37], which characterizes the minimal constant that bounds the local variation of a function, to simplify:

$$|\xi_{(m)} - \xi'_{(m)}| \leq \tau_{(m)} \|x_i^{(m)} - x_i^{(m)'}\|_2 \quad (5)$$

where $\xi'_{(m)}$ denotes the UCoM of the sample $x_i^{(m)}$ after being perturbed $x_i^{(m)'}$. Finally, we can provide the lower bound of multimodal robustness in late fusion framework. The proof details are provided in the Appendix A.1.

**Theorem 3.1.** *Multimodal Robustness of the Late Fusion Multimodal System. Given an input $x_i$ and a perturbed sample $x_i'$, for the target class $y$ and the closest competing class $j \neq y$, let $\xi_{(m)}$ denote the UCoM for the $m$-th modality under a Lipschitz constraint $\tau_{(m)}$. Let $w^{(m)}$ represent the weight assigned to the $m$-th modality in a late fusion scheme. The lower bound of the perturbation radius in the late fusion framework when $|\mathcal{M}| = 2$ can then be described as:*

$$R(x_i) = \min \|x_i - x_i'\|_2 \geq \frac{w^{(1)}\xi_{(1)} + w^{(2)}\xi_{(2)}}{\sqrt{(w^{(1)}\tau_{(1)})^2 + (w^{(2)}\tau_{(2)})^2}}. \quad (6)$$

**Corollary 3.2.** *Larger Unimodal Confidence Margins lead to greater robustness in multimodal systems.*

### 3.3 Multimodal Negative Learning

In multimodal tasks, significant disparities in modality quality and information capacity are common. Weak modalities often struggle to produce accurate predictions independently, especially as inter-modal imbalance increases, leading to greater uncertainty in the output space. Intuitively, it is easier for such modalities to suppress incorrect predictions than to identify the correct one. Thus, leveraging high-confidence predictions from the dominant modality to suppress uncertainty in the weak modality over non-target classes is a natural choice. On one hand, a dominant modality that is more accurate on the target class typically exhibits lower uncertainty on non-target classes, enabling it to denoise the weak modality and enhance cross-modal consistency. On the other hand, restricting the guidance to non-target classes helps prevent the weak modality from over-aligning with the dominant one, thereby preserving its complementary information rather than being overwhelmed. Further details on preserving unique information of the weaker modality are provided in Appendix B.1.

However, Theorem 3.1 and Corollary 3.2 show that relying only on confidence in the ground-truth class to define modality roles can be risky. Guiding the weak modality using non-target signals from the strong one may reduce its margin and harm robustness. To avoid this, we redefine dominant and inferior modalities to ensure both lower uncertainty and preserved robustness.

**Definition 3.3.** A modality exhibiting higher confidence in the target class and a larger UCoM is regarded as the **Robust Dominant Modality (RDM)**; the others are considered the **Inferior Modality (IM)**.

Therefore, when the robust dominant modality exhibits lower uncertainty over non-target classes and a larger UCoM, guiding the inferior modality using the non-target information from the dominant enables the inferior modality to eliminate more uncertain choices and enlarge its own margin. This ultimately enhances the overall robustness of the multimodal system. Based on this intuition, we propose Multimodal Negative Learning:

$$MNL(P^{(RDM)}, P^{(IM)}, \overline{y}) = -\overline{y} \cdot P^{(RDM)} \cdot \log(P^{(IM)}) \quad (7)$$

where $P^{(m)} = \sigma(f^{(m)}(x_i^{(m)}))$, $\sigma(\cdot)$ denotes the softmax function and $\overline{y}$ indicates 0 at the ground-truth class and 1 for all non-target classes. Most importantly, during the optimization of MNL, we detach the predictions from the robust dominant modality. Furthermore, an analysis of the MNL from the perspective of empirical error reduction is provided in the Appendix A.2.

### 3.4 Dynamic Guidance and Training Strategy

It is important to note that modality dominance is not a fixed property. Instead, it emerges as a dynamic phenomenon that varies across samples, tasks, and training iterations. Therefore, we implement dynamic guidance between modalities based on the variation in modality predictions. Specifically, when $|\mathcal{M}| = 2$:

$$MNL(P^{(RDM)}, P^{(IM)}, \overline{y}) = \begin{cases} -\overline{y} \cdot P^{(1)} \cdot \log(P^{(2)}) & P_y^{(2)} < P_y^{(1)}, \xi_{(2)} < \xi_{(1)} \\ -\overline{y} \cdot P^{(2)} \cdot \log(P^{(1)}) & P_y^{(1)} < P_y^{(2)}, \xi_{(1)} < \xi_{(2)} \end{cases} \tag{8}$$

The dynamic guidance mechanism is a critical component of MNL. Its core is formalized in Definition 3.3, which jointly considers both $P_y$ and UCoM to identify the RDM and the IMs. If dynamic guidance is absent in multimodal negative learning, an incorrect modality may guide a correct one, and a low UCoM modality might influence a high UCoM one, potentially weakening the multimodal system's robustness. Moreover, the motivation behind the MNL is to reduce the uncertainty of the inferior modality, thereby enabling it to better focus on the correct answer. Therefore, in addition to negative learning for the non-target classes, it is also intuitive to positive learning for the target class. For the target class, the standard cross-entropy loss naturally fulfills this objective. Consequently, in our training process, the overall loss is defined as:

$$\mathcal{L} = CE(P^{fusion}, y) + \sum_{i=1}^{2} CE(P^{(i)}, y) + \lambda \cdot MNL(P^{(RDM)}, P^{(IM)}, \overline{y}) \tag{9}$$

where $CE(\cdot, \cdot)$ denotes the cross-entropy loss, $P^{fusion} = \sigma(f(x))$. $y$ is the one-hot label, which equals 1 at the ground truth class. The hyperparameter $\lambda$ controls the strength of MNL. The training process is divided into two stages. During Stage 1, we employ only the $CE(\cdot, \cdot)$ loss to optimize for the target class. In Stage 2, once the performance of both modalities stabilizes, we incorporate the MNL to better exploit the guidance from the dominant modality to the inferior modality. The warm-up setting primarily serves to reduce unnecessary computational cost in the early phase of MNL training. The overall pseudocode of our model is provided in the Appendix C.6.

## 4 Experiments

### 4.1 Experimental Setup

**Datasets.** We evaluate the proposed method on a variety of multimodal classification tasks, including (1) image-text classification using the UMPC Food-101 dataset [38], which contains approximately 100,000 image-text recipe pairs collected in uncontrolled environments, covering 101 food categories with varying levels of noise in both modalities; (2) sentiment analysis using the MVSA dataset [39], which consists of paired user-generated images and texts annotated with sentiment polarity; (3) scene recognition based on the NYU Depth V2 dataset [40], which includes RGB-depth image pairs primarily focused on indoor scene understanding; and (4) emotion recognition using the CREMA-D dataset [41], a multimodal audio-visual dataset in which actors express six basic emotions (happiness, sadness, anger, fear, disgust, neutrality) through spoken utterances.

**Evaluation metrics.** To ensure consistency with prior work [33, 42, 43], we evaluate average model performance under Gaussian and Salt noise for image modality, SNR-based noise for audio modality, and blank noise for the text modality. To reduce the variance introduced by stochastic factors, we conduct evaluations using five independent random seeds.

**Baselines and competing methods.** In our experiments, we integrate the proposed MNL with both static late fusion method (LATE FUSION, LF) and advanced dynamic late fusion approaches. Specifically, we compare our method against representative static fusion strategies as well as state-of-the-art dynamic fusion techniques such as DynMM [44], TMC [43], QMF [42], and PDF [33]. In addition, we establish unimodal baselines to provide a comprehensive evaluation.

**Implementation details.** The network is trained using the original configuration of the corresponding method, with the MNL directly incorporated. All experiments are conducted on an NVIDIA TITAN GPU using PyTorch, with default settings applied across all methods. The warm-up time of Stage 1 and additional details are provided in the Appendix C.

Table 1: Performance of different methods under varying noise levels across four (MVSA, FOOD101, NYU Depth V2, and CREMA-D) datasets. We add noise (**Gaussian noise** for image-related modalities) on 50% modalities and $\varepsilon$ presents the noise degree. The red and blue represent the best and second-best result respectively. We used ↑ and ↓ to illustrate the amount of increase or decrease our MNL method achieved. LF denotes static late fusion. Full results with standard deviation are reported in Appendix 14.

| Method | MVSA | | | UMPC FOOD 101 | | | NYU DEPTH V2 | | | CREMA-D | | |
|---|---|---|---|---|---|---|---|---|---|---|---|---|
| | $\varepsilon$=0.0 | 5.0 | 10.0 | $\varepsilon$=0.0 | 5.0 | 10.0 | $\varepsilon$=0.0 | 5.0 | 10.0 | $\varepsilon$=0.0 | 5.0 | 10.0 |
| Unimodal1 | 64.12 | 49.36 | 45.00 | 64.62 | 34.72 | 33.03 | 63.30 | 53.12 | 45.46 | 60.70 | 59.60 | 49.52 |
| Unimodal2 | 75.61 | 69.50 | 47.41 | 86.46 | 67.38 | 43.88 | 62.65 | 50.95 | 44.13 | 56.23 | 52.47 | 38.17 |
| LF | 76.88 | 63.46 | 55.16 | 90.69 | 68.49 | 57.99 | 70.03 | 64.37 | 60.55 | 68.04 | 64.25 | 52.39 |
| DYNMM | 79.07 | 67.96 | 59.21 | 92.59 | 74.74 | 59.68 | 65.50 | 54.31 | 46.79 | 63.27 | 62.01 | 51.43 |
| TMC | 74.87 | 66.72 | 60.35 | 89.86 | 73.93 | 61.37 | 70.40 | 59.33 | 50.61 | 63.63 | 62.68 | 57.97 |
| QMF | 78.07 | 73.85 | 61.28 | 92.90 | 76.03 | 62.21 | 69.54 | 64.10 | 60.18 | 66.13 | 64.27 | 50.77 |
| PDF | 79.94 | 74.40 | 63.09 | 93.32 | 76.47 | 62.83 | 71.37 | 65.72 | 62.56 | 67.07 | 64.57 | 53.33 |
| LF+**MNL** | 79.50 | 74.03 | 63.01 | 92.77 | 75.16 | 62.06 | 71.05 | 67.02 | 63.81 | 73.71 | 70.35 | 57.26 |
| Δ | ↑2.62 | ↑10.57 | ↑7.85 | ↑2.08 | ↑6.67 | ↑4.06 | ↑1.02 | ↑2.65 | ↑3.26 | ↑5.67 | ↑6.10 | ↑4.87 |
| QMF+**MNL** | 79.45 | 74.12 | 62.75 | 93.03 | 75.41 | 62.59 | 71.25 | 65.38 | 61.80 | 68.18 | 67.00 | 52.62 |
| Δ | ↑1.38 | ↑0.27 | ↑1.47 | ↑0.13 | ↓0.62 | ↑0.38 | ↑1.71 | ↑1.28 | ↑1.62 | ↑2.05 | ↑2.73 | ↑1.85 |
| PDF+**MNL** | 80.54 | 74.07 | 63.78 | 93.33 | 76.65 | 63.16 | 71.52 | 67.01 | 63.07 | 69.18 | 66.94 | 55.43 |
| Δ | ↑0.60 | ↓0.33 | ↑0.69 | ↑0.01 | ↑0.18 | ↑0.33 | ↑0.15 | ↑1.29 | ↑0.51 | ↑2.11 | ↑2.37 | ↑2.10 |

## 4.2 Results

Tables 1 and 2 present the main results across all datasets. For the MVSA and FOOD-101 datasets, Unimodal1 and Unimodal2 refer to the image and text modalities, respectively. In the NYU Depth V2 dataset, they correspond to depth and RGB, while in the CREMA-D dataset, they denote the audio and visual modalities. Specifically, Gaussian noise is added to image-related modalities in Table 1, Salt noise is added to image-related modalities in Table 2, blank (masking) noise is applied to text, and noise is added to audio by adjusting the signal-to-noise ratio (SNR) according to the noise level following [33, 42]. We standardize the noise levels across modalities as $\epsilon = 0, 5$, and 10, allowing us to evaluate the robustness of multimodal models under varying degrees of noise. MNL enhances the model's robustness compared to the baselines and competing methods. At multiple noise levels, MNL significantly improves the performance of the static late fusion method. In most cases, MNL consistently outperforms both the PDF and QMF baselines.

Notably, the improvement brought by MNL is less pronounced when applied to dynamic fusion methods compared to static fusion strategy. As indicated in Theorem 3.1 and Corollary 3.2, the lower bound of multimodal robustness is jointly influenced by both the modality weights and the UCoM. Dynamic fusion strategies [33, 42] generally assign higher weights to modalities that are more confident in predicting the target class, while overlooking the potential synergy between modality weighting and UCoM enhancement. Specifically, MNL encourages weaker modalities to increase their margins, but dynamic fusion tends to down-weight these weaker modalities, resulting in a misalignment. This misalignment can diminish the benefits of MNL in the dynamic fusion setting and may even lead to performance degradation relative to the baseline. In most cases, increasing the UCoM of the inferior modality leads to higher confidence in predicting the ground truth. On the NYU Depth V2 dataset, MNL improves the performance of both static and dynamic fusion methods. However, the gains are relatively modest, likely because the dataset exhibits minimal imbalance between the two modalities. In other words, the gap between the robust dominant and inferior modalities is small. As a result, the dominant modality has limited capacity to guide the inferior modality, reducing the effectiveness of the margin enhancement encouraged by MNL.

## 4.3 Ablation Study

Table 3 compares the performance of three guidance strategies—Prior, Confident, and Robust—under different noise levels ($\epsilon = 0, 5, 10$) on both dynamic PDF and static late fusion frameworks. Prior

Table 2: Performance of different methods under varying noise levels across four (MVSA, FOOD101, NYU Depth V2, and CREMA-D) datasets. We add noise (**Salt noise** for image-related modalities) on 50% modalities and $\varepsilon$ presents the noise degree. LF denotes static late fusion. Full results with standard deviation are reported in Appendix 15.

| Method | MVSA | | | UMPC FOOD101 | | | NYU Depth V2 | | | CREMA-D | | |
|---|---|---|---|---|---|---|---|---|---|---|---|---|
| | $\varepsilon$=0.0 | 5.0 | 10.0 | $\varepsilon$=0.0 | 5.0 | 10.0 | $\varepsilon$=0.0 | 5.0 | 10.0 | $\varepsilon$=0.0 | 5.0 | 10.0 |
| Unimodal1 | 64.12 | 56.72 | 50.71 | 64.62 | 50.75 | 36.83 | 63.30 | 50.99 | 38.56 | 60.70 | 59.60 | 49.52 |
| Unimodal2 | 75.61 | 69.50 | 47.41 | 86.46 | 67.48 | 43.88 | 62.65 | 49.14 | 34.76 | 56.23 | 50.62 | 43.90 |
| LF | 76.88 | 67.88 | 55.43 | 90.69 | 77.99 | 58.75 | 70.03 | 62.05 | 51.50 | 68.04 | 62.50 | 52.61 |
| DYNMM | 79.07 | 71.35 | 59.96 | 92.59 | 78.91 | 57.64 | 65.50 | 52.26 | 38.17 | 63.27 | 62.92 | 52.33 |
| TMC | 74.87 | 68.02 | 56.62 | 89.86 | 77.86 | 60.22 | 70.40 | 59.33 | 45.32 | 63.63 | 62.31 | 58.44 |
| QMF | 78.07 | 73.90 | 60.41 | 92.90 | 80.87 | 61.60 | 69.54 | 62.02 | 51.87 | 66.13 | 63.73 | 51.55 |
| PDF | 79.94 | 75.11 | 61.97 | 93.32 | 81.21 | 61.76 | 71.37 | 64.27 | 53.62 | 67.07 | 63.44 | 53.71 |
| LF+**MNL** | 79.50 | 74.68 | 62.31 | 92.77 | 80.87 | 61.41 | 71.05 | 64.88 | 54.08 | 73.71 | 68.06 | 58.41 |
| Δ | ↑2.62 | ↑6.80 | ↑6.88 | ↑2.08 | ↑2.88 | ↑2.66 | ↑1.02 | ↑2.83 | ↑2.58 | ↑5.67 | ↑5.56 | ↑5.80 |
| QMF+**MNL** | 79.45 | 75.14 | 64.68 | 93.03 | 81.14 | 62.47 | 71.25 | 63.62 | 53.30 | 68.18 | 64.05 | 52.32 |
| Δ | ↑1.38 | ↑1.24 | ↑4.27 | ↑0.13 | ↑0.27 | ↑0.87 | ↑1.71 | ↑1.60 | ↑1.43 | ↑2.05 | ↑0.32 | ↑0.77 |
| PDF+**MNL** | 80.54 | 75.76 | 64.93 | 93.33 | 81.52 | 62.95 | 71.52 | 64.32 | 54.08 | 69.18 | 63.77 | 54.30 |
| Δ | ↑0.60 | ↑0.65 | ↑2.96 | ↑0.01 | ↑0.31 | ↑1.19 | ↑0.15 | ↑0.05 | ↑0.46 | ↑2.11 | ↑0.33 | ↑0.59 |

uses fixed modality dominance from prior knowledge, Confident dynamically selects guidance based on each modality's confidence, and Robust incorporates UCoM into dynamic guidance.

Our proposed MNL corresponds to the Confident + Robust (✓✓) setting, which jointly considers modality-wise confidence and UCoM to dynamically determine the guidance direction. This dynamic guidance mechanism not only identifies which modality is currently more trustworthy, but also detects and compensates for potential modality imbalance, enhancing fusion stability. As shown in the results, our method consistently outperforms all other variants. For instance, on PDF at $\epsilon = 10$, it achieves 63.78%, compared to 63.24% for Confident-only and 61.02% for Prior-only. Similar trends are observed on LATE FUSION, with our method reaching 63.01%, significantly better than Prior-only (62.77%) and Confident-only (59.35%). These results demonstrate that our joint Confident + Robust guidance leads to more adaptive and balanced decision fusion, especially under noisy and imbalanced conditions where single-criterion strategies fail to generalize. In general, benefiting from the dynamic guidance mechanism, MNL can improve the performance of most methods and achieve optimal results.

Table 4 presents an ablation study comparing two guidance strategies: All-Class guidance, which aligns inferior modalities across the full label space, and our proposed Non-Target guidance, which focuses only on suppressing non-target categories. Across all perturbation levels, Non-Target guidance consistently outperforms All-Class guidance in both the dynamic PDF framework and the static LATE FUSION baseline. For instance, under the highest noise level ($\epsilon$ = 10), Non-Target guidance improves accuracy from 61.56% to 63.78 % on PDF (+2.22%) and from 62.52% to 63.01% on LATE FUSION (+0.49%). Even in the clean setting ($\epsilon$ = 0), it shows noticeable gains: +1.9% for PDF and +0.60% for LATE

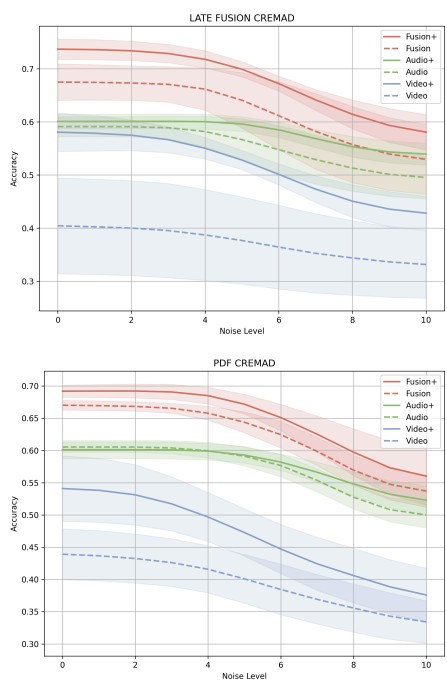

Figure 3: Accuracy varies with Gaussian noise level on the CREMA-D dataset. "+" indicates models trained with the proposed Multimodal Negative Learning (MNL). Our method consistently improves accuracy across all noise levels.

Table 3: Performance comparison under different guidance strategies on MVSA dataset.

| Prior | Confident | Robust | $\epsilon = 0$ | $\epsilon = 5$ | $\epsilon = 10$ |
|---|---|---|---|---|---|
| LATE FUSION | | | 76.88 | 63.46 | 55.16 |
| ✓ | | | 78.66 | 72.69 | 62.77 |
| | ✓ | | 78.74 | 71.87 | 59.35 |
| | ✓ | ✓ | 79.50 | 74.03 | 63.01 |
| PDF | | | 79.94 | 74.40 | 63.09 |
| ✓ | | | 79.19 | 71.85 | 61.02 |
| | ✓ | | 80.23 | 72.68 | 63.24 |
| | ✓ | ✓ | 80.54 | 74.07 | 63.78 |

Table 4: Performance comparison under all-class and non-target guidance across different perturbation levels on MVSA dataset.

| All-Class | Non-Target | $\epsilon = 0$ | $\epsilon = 5$ | $\epsilon = 10$ |
|---|---|---|---|---|
| LATE FUSION | | 76.88 | 63.46 | 55.16 |
| ✓ | | 78.90 | 72.16 | 62.52 |
| | ✓ | 79.50 | 74.03 | 63.01 |
| PDF | | 79.94 | 74.40 | 63.09 |
| ✓ | | 78.64 | 72.45 | 61.56 |
| | ✓ | 80.54 | 74.07 | 63.78 |

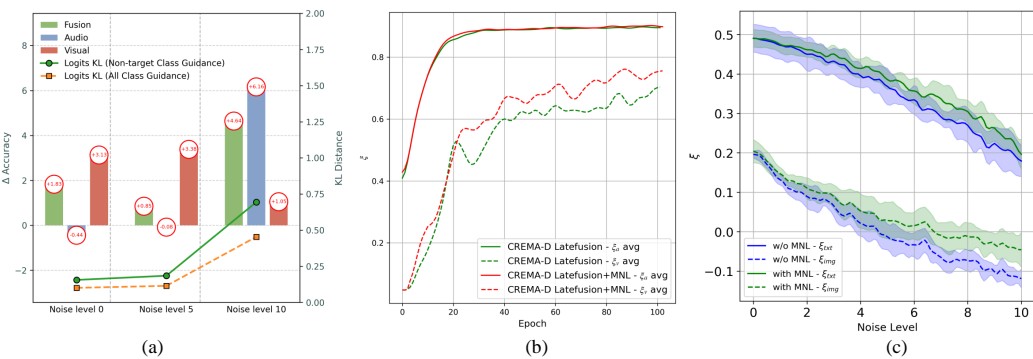

(a)     (b)     (c)

Figure 4: **(a)** We visualize the average KL divergence between the two modality predictions and compare the performance gaps between our method and the baseline across varying noise levels. **(b)** and **(c)** show the evolution of UCoM on the CREMA-D and MVSA datasets, respectively.

FUSION. These consistent improvements verify that Non-Target guidance leads to better noise resilience and decision stability, by avoiding overfitting to uncertain target predictions and instead stabilizing the decision space through suppressing irrelevant classes. Further details regarding the settings of $\lambda$ and the warm-up epochs can be found in Appendix D.2.

### 4.4 Discussion

**Analyzing the Sources of Performance Improvements.** Fig. 3 illustrates the performance of each modality and fusion model under increasing noise levels ($\epsilon = 0$–$10$), with and without the proposed MNL loss. We observe that adding the MNL loss ("+" curves) consistently improves accuracy across all settings. For example, under the static late fusion setting (top), Fusion+ outperforms Fusion by up to 3.2% at $\epsilon = 6$, and maintains higher robustness as noise increases. Similarly, in the dynamic PDF setting (down), Fusion+ indicates that MNL not only improves clean performance but also significantly reduces performance degradation under perturbation. Moreover, both Audio+ and Video+ curves show that MNL benefits single-modality branches as well, with especially noticeable gains on the weak video modality. This confirms that MNL effectively guides weak modalities to suppress noisy predictions, stabilizing decision fusion and mitigating modality imbalance.

**Analysis of Guidance on Non-Target Classes.** Fig. 4(a) illustrates the optimization behavior of MNL by comparing the average KL divergence between dual-modal outputs and the corresponding performance gap under different guidance strategies. We use the average KL divergence between modality predictions to quantify modality discrepancy, and compare our Non-Target class guidance with All-Class guidance under varying noise levels. Results show that although Non-Target guidance leads to higher KL divergence, it consistently achieves better multimodal performance across three noise levels, with gains of +1.83%, +0.85%, and +4.64%, respectively. Notably, the weaker modality benefits the most from this strategy. This seemingly counterintuitive result highlights a core strength of our method: unlike All-Class guidance, which enforces rigid alignment by pushing the weaker modality to imitate the stronger one, our Non-Target guidance selectively preserves discriminative cues in the weaker modality. This selective preservation maintains modality diversity and enhances cross-modal complementarity, ultimately improving system robustness.

**Analysis of the Unimodal Confidence Margin.** For presentation, UCoM is normalized via a softmax over the output logits. Fig. 4(b) shows the UCoM of correctly classified validation samples during training for LATE FUSION and LATE FUSION+MNL on CREMA-D. Typically, one modality quickly stabilizes while the weaker modality lags at a lower UCoM. As UCoM directly governs multimodal robustness (Theorem 3.1), this gap is critical. MNL consistently boosts the weak modality's UCoM, reinforcing its decision boundary and improving overall robustness to perturbations. And Fig. 4(c) shows the $\xi$ values for all samples of each modality under different noise levels on the MVSA dataset. Notably, for weak modality (e.g., image), $\xi$ is consistently higher with MNL, which ultimately leads to improved model performance under noisy conditions.

Furthermore, theoretical analysis reveals that the effectiveness of MNL depends on satisfying a robustness condition. As shown in a representative case, when modality A exhibits a higher predicted probability than modality B, $P_y^{(A)} > P_y^{(B)}$ but a lower UCoM $\xi_{(A)} < \xi_{(B)}$, directly using the non-target class signals from A modality to guide the B modality can actually reduce the UCoM of B. This phenomenon highlights that guidance strategies lacking robustness-aware constraints can disrupt UCoM, ultimately degrading overall system performance and robustness.

**Analysis of the extensibility of MNL.** MNL demonstrates strong scalability. In terms of the number of modalities, according to Definition 3.3, it can be easily extended to multiple modalities. We conducted experiments on the CMU-MOSEI dataset [34] under varying levels of noise interference, including the visual, textual, and audio modalities in Table 5. MNL consistently improves model performance across both static and dynamic fusion methods, particularly under noisy conditions. Moreover, MNL can be further extended to LLMs QA tasks and MLLMs VQA tasks. Specifically, we apply logit fusion to the answer token generated by two models, a process we refer to as LATE FUSION. During optimization, MNL enables the more robust and accurate model to guide the weaker model in eliminating incorrect answers, given that the models share the same vocabulary. We conducted experiments on the MathQA dataset [45] (QA task). For the QA task, we fine-tuned Qwen2.5-0.5B-Instruct and Qwen2.5-1.5B [46] using LoRA [47]. We then compared the performance to validate the effectiveness of our approach in Table 6. Further details about the task setting and the VQA task are provided in Appendix C.4.

Table 5: Results of experiments conducted on the CMU-MOSEI dataset.

| Method | $\epsilon = 0$ | $\epsilon = 5$ | $\epsilon = 10$ |
|---|---|---|---|
| LATE FUSION | 66.42 | 61.71 | 45.80 |
| + MNL | 67.44 | 63.36 | 55.09 |
| PDF | 66.14 | 63.54 | 42.47 |
| + MNL | 67.29 | 64.35 | 48.62 |

Table 6: Results of the QA task on MathQA dataset, where M1 is Qwen2.5-0.5B-Instruct and M2 is Qwen2.5-1.5B.

| Method | Fusion | M1 | M2 |
|---|---|---|---|
| LATE FUSION | 50.89 | 42.85 | 49.41 |
| + MNL | 51.42 | 43.32 | 50.85 |

## 5   Conclusion

In this work, we address the challenge of modality imbalance and fragility in multimodal fusion by rethinking the role of weak modalities. Rather than forcing weak modalities to align with strong ones, we propose a novel Multimodal Negative Learning (MNL) framework that guides weak modalities to suppress non-target classes while preserving their unique contributions. Theoretically, we prove that this strategy tightens the robustness lower bound in decision-level fusion and reduces the empirical error of weak modalities. Our method can be flexibly integrated into existing late fusion architectures. Extensive experiments on noisy and imbalanced benchmarks demonstrate consistent improvements in both accuracy and robustness, confirming the effectiveness of our approach.

In future work, we plan to extend this framework to more complex multimodal scenarios, such as multi-label classification, open-set recognition, and temporal/sequential fusion. In addition, we are interested in developing more fine-grained uncertainty estimation methods to enhance dynamic guidance, and in exploring theoretical generalization bounds under adversarial or missing-modality settings. We believe that our perspective offers a new and promising direction for building more robust, adaptive, and trustworthy multimodal systems.

## Acknowledgements

This work was supported by the National Natural Science Foundation of China (Nos. 62222608, 62476198, U23B2049, 62436002), the National Natural Science Foundation of China (Grant No. 22527901) through the National Major Research Instrumentation Program, the Fundamental and Interdisciplinary Disciplines Breakthrough Plan of the Ministry of Education of China (JYB2025XDXM503), the Natural Science Foundation of Tianjin (No. 25JCYBJC00950), and the CCF-Baidu Open Fund. The authors thank the anonymous reviewers for their helpful remarks.

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

# Appendix

## A Proof

### A.1 Robust Lower Bounds for Multimodal Learning with Late Fusion

In the decision-level late fusion framework, following the standard setting, let $f^{(m)}$ denote the logit output from modality $m$, where $C$ is the number of classes and $B$ is the batch size. For simplicity, we illustrate the formulation using two modalities. The fusion weights satisfy $w_1 + w_2 = 1$, with $w_1, w_2 > 0$. The fused multimodal output is defined as:

$$f(x) = w^{(1)} \cdot f^{(1)}(x^{(1)}) + w^{(2)} \cdot f^{(2)}(x^{(2)}) \tag{10}$$

where $x = \{x^{(1)}, x^{(2)}\}$, $f(x) \in \mathbb{R}^{B \times C}$. According to the definition of UCoM, for a given ground-truth class $y$ and competing class $k \neq y$ (the runner-up class), the UCoM is given by:

$$\xi_{(m)} = f^{(m)}(x^{(m)})_y - f^{(m)}(x^{(m)})_k \tag{11}$$

In real-world scenarios, modality inputs are often subject to perturbations. Let the perturbed input be denoted as $x' = \{x^{(1)'}, x^{(2)'}\}$. We consider a critical condition under which the perturbed multimodal margin becomes zero for some $k \neq y$,

$$f(x')_y - f(x')_k = \sum_{m=1}^{2} w^{(m)} \cdot \xi'_{(m)} = \sum_{m=1}^{2} w^{(m)} \cdot (f^{(m)}(x^{(m)'})_y - f^{(m)}(x^{(m)'})_k) = 0 \tag{12}$$

The perturbed multimodal logit output can then be expressed as:

$$f(x') = w^{(1)} \cdot f^{(1)}(x^{(1)'}) + w^{(2)} \cdot f^{(2)}(x^{(2)'}) \tag{13}$$

where $x' = \{x^{(1)'}, x^{(2)'}\}$, $f(x') \in \mathbb{R}^{B \times C}$. We now compute the multimodal CoM between class $y$ and the competing class $k$:

$$f(x)_y - f(x)_k = w^{(1)} \cdot \xi_{(1)} + w^{(2)} \cdot \xi_{(2)} - \left( w^{(1)} \cdot \xi'_{(1)} + w^{(2)} \cdot \xi'_{(2)} \right) \tag{14}$$

$$= w^{(1)}(\xi_{(1)} - \xi'_{(1)}) + w^{(2)}(\xi_{(2)} - \xi'_{(2)}). \tag{15}$$

According to the Lipschitz continuity assumption [16, 37], for modality $m$, there exists a constant $\tau_{(m)}$ such that:

$$|\xi_{(m)} - \xi'_{(m)}| \leq \tau_{(m)} \|x^{(m)} - x'^{(m)}\|_2. \tag{16}$$

Combining Equation 15 and Equation 16, the total margin variation is upper bounded as:

$$f(x)_y - f(x)_k \leq w^{(1)} \tau_{(1)} \|x^{(1)} - x^{(1)'}\|_2 + w^{(2)} \tau_{(2)} \|x^{(2)} - x^{(2)'}\|_2. \tag{17}$$

Using the Cauchy-Schwarz inequality, define

$$a = w^{(1)} \tau_{(1)}, \quad b = w^{(2)} \tau_{(2)},$$
$$u = (a, b), \quad v = (\|x^{(1)} - x^{(1)'}\|_2, \|x^{(2)} - x^{(2)'}\|_2),$$

then

$$\langle u, v \rangle \leq \|u\|_2 \cdot \|v\|_2 = \sqrt{a^2 + b^2} \cdot \|x - x'\|_2. \tag{18}$$

Therefore,

$$f(x)_y - f(x)_k \leq \sqrt{(w^{(1)} \tau_{(1)})^2 + (w^{(2)} \tau_{(2)})^2} \cdot \|x - x'\|_2. \tag{19}$$

Hence, the multimodal robustness radius is lower bounded as:

$$R(x) = \min \|x - x'\|_2 \geq \frac{w^{(1)} \xi_{(1)} + w^{(2)} \xi_{(2)}}{\sqrt{(w^{(1)} \tau_{(1)})^2 + (w^{(2)} \tau_{(2)})^2}}. \tag{20}$$

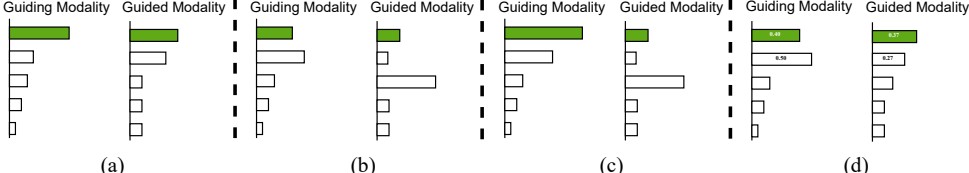

Figure 5: During the training process, examples are provided for both the guiding and guided modalities, with the green bar representing the ground truth class. Specifically, (d) highlights the predicted probabilities for the ground truth class and the runner-up class.

## A.2 Empirical Error

We incorporate Multimodal Negative Learning (MNL) into both traditional static late fusion methods and advanced dynamic fusion approaches. Within the MNL framework, we define one modality as the *guiding modality* (Robust Dominant Modality, RDM) and the other as the *guided modality* (Inferior Modality, IM). To analyze the impact of MNL on empirical error, we conduct a case study based on different combinations of prediction correctness:

1. **Both modalities predict correctly or both incorrectly**: Since MNL operates independently of the ground-truth label, it does not directly affect the empirical error in these cases, as illustrated in Fig. 5(a) and (b).

2. **The guiding modality predicts correctly, while the guided modality predicts incorrectly**: As shown in Fig. 5(c), MNL encourages the guided modality to align with the guiding modality on non-target classes. This alignment helps steer the guided modality toward the correct prediction, thereby reducing the empirical error.

3. **The guiding modality predicts incorrectly, while the guided modality predicts correctly**: As illustrated in Fig. 5(d), from the perspective of empirical error, if the guiding and guided roles are assigned solely based on prediction confidence, it is possible for the guiding modality to have a higher confidence score on the ground truth class ($P_{\text{guiding}} > P_{\text{guided}}$) but a lower UCoM ($\xi_{\text{guiding}} < \xi_{\text{guided}}$). In such cases, the guiding modality may be making an incorrect prediction while the guided modality is actually correct. Therefore, in MNL, the assignment of guiding and guided roles is not solely based on prediction confidence or the probability of the ground truth class. Instead, it also considers the UCoM. As a result, instances like the one shown in Fig. 5(d), where the guiding modality is incorrect despite high confidence, are excluded from contributing to the guidance.

Overall, integrating MNL into existing late fusion methods leads to a consistent reduction in empirical error.

## B More Analysis

### B.1 Analysis of Unique Information in Modalities

In this subsection, we demonstrate the effectiveness of our approach from a mutual information perspective. Let the two modalities be represented by $\mathcal{X}_1$ and $\mathcal{X}_2$, and let $\mathcal{Y}$ denote the ground-truth labels. Following the methodology in [48, 49], we decompose the multimodal mutual information $I(\mathcal{X}_1, \mathcal{X}_2; \mathcal{Y})$ into three conditional mutual information components:

$$I(\mathcal{X}_1, \mathcal{X}_2; \mathcal{Y}) = \underbrace{I(\mathcal{X}_1; \mathcal{X}_2; \mathcal{Y})}_{S(\mathcal{X}_1, \mathcal{X}_2) = \text{relevant shared info.}} + \underbrace{I(\mathcal{X}_1, \mathcal{Y}|\mathcal{X}_2)}_{U(\mathcal{X}_1) = \text{relevant unique info. in } \mathcal{X}_1} + \underbrace{I(\mathcal{X}_2, \mathcal{Y}|\mathcal{X}_1)}_{U(\mathcal{X}_2) = \text{relevant unique info. in } \mathcal{X}_2}$$

In the main paper, we analyze the relationship between the KL divergence of modality outputs and overall system performance to investigate the unique role of non-target class guidance in multimodal learning. Unlike traditional all-class guidance, the non-target strategy adopted by MNL does not enforce strict alignment across modalities. Instead, it selectively preserves discriminative features in the weaker modality. In other words, MNL's strength lies in the fact that the dominant modality is

more effective at excluding non-target classes, while the weaker modality can learn this exclusion ability without compromising its own distinctive representation for target classes. This prevents the dominant modality from overwhelming the weaker one and helps maintain a healthy diversity between modalities.

Specifically, we employ the *dit* toolkit [50] to perform partial information decomposition, quantifying the unique information contributed by each modality. It is important to note that this quantification reflects only the relative magnitudes of unique information within the multimodal system.

Fig. 6 reports the relative unique information of each modality under varying noise levels on the CREMA-D dataset, measured as the ratio of unique information from the weaker modality to that from the stronger one. Under zero-noise conditions, methods that apply guidance across all classes tend to suppress the contribution of the weaker modality, resulting in limited benefits for fusion. In contrast, MNL, which applies guidance only to non-target classes, consistently reduces the disparity between modalities and effectively preserves or enhances the unique information of the weaker modality.

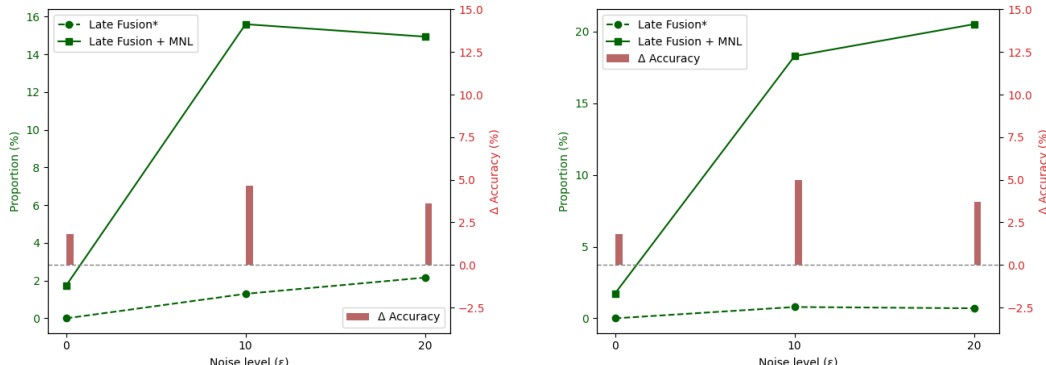

Figure 6: Results on the CREMA-D dataset under Gaussian noise (**left**) and Salt noise (**right**). "*" denotes methods applying guidance across all classes. The lines show the relative modality-specific unique information (**Proportion**), measured as the ratio of the weaker to the stronger modality. ΔAccuracy reflects the performance improvement brought by the MNL method over conventional all-class guidance approaches.

## B.2   Extended Training for Improved Audio Modality Performance

In the main paper, based on previous methods, we compared the multimodal and unimodal performances of two strategies, Non-target Class Guidance (MNL) and All Class Guidance, after training for 100 epochs under different noise levels. The results show that at $\epsilon = 0$ and $\epsilon = 5$, the audio modality under the MNL method performs worse than that under All Class Guidance. This may be due to the dynamic guidance mechanism of MNL causing insufficient training of the audio modality. To this end, we conducted two separate experiments: one extending the training to 200 epochs, and the other increasing the loss weight of the audio modality from 1.0 to 2.0.

Fig. 7a and 7b illustrate the performance differences between the MNL method and the All-Class Guidance baseline under various levels of Gaussian noise, with extended training duration and increased loss weight for the audio modality, respectively. Notably, MNL continues to achieve superior multimodal fusion performance, yielding improvements of (+0.83%, +1.14%, +6.45%) after training for 200 epochs, and (+1.88%, +1.20%, +5.58%) when the loss weight for the audio modality is increased to 2.0.

More importantly, MNL not only improves the overall multimodal performance but also consistently outperforms All-Class Guidance in the audio modality alone, with gains of (+0.23%, +0.13%, +2.99%) in Fig. 7a, and (+0.77%, +0.10%, +5.81%) in Fig. 7b. These results indicate that prolonged training or accelerating the learning of the stronger modality can be beneficial when adopting the MNL strategy.

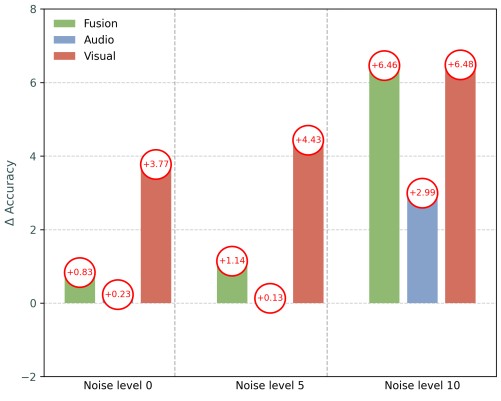
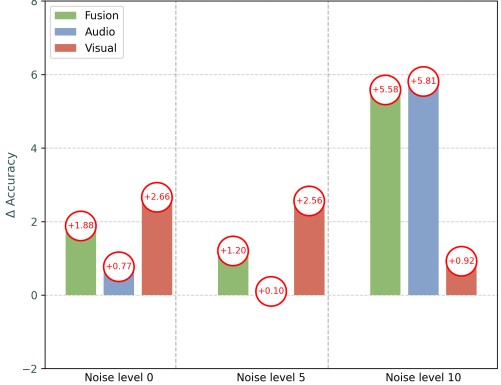

(a) After training for 200 epochs    (b) Increase the loss weight of the audio modality

Figure 7: On the CREMA-D dataset with added Gaussian noise, we compared the performance differences between the MNL method and the approach guided across all class under various noise levels.

## B.3   Advantages of Late Fusion and Comparison with Latent Fusion

In multimodal learning, the advantages of late fusion lie in its **flexibility**, **robustness**, and **interpretability** [36].

- **Flexibility.** Latent fusion requires aligning and projecting heterogeneous modalities into a unified representation space, which is often challenging and prone to noise or information loss. In contrast, late fusion processes each modality independently and combines only the final predictions, significantly reducing the need for strict alignment. As a result, late fusion is more flexible and better suited for integrating diverse multimodal data types.

- **Robustness.** In real-world scenarios, data from one modality may be missing entirely or severely degraded (e.g., due to noise). In latent fusion, such incomplete or low-quality modalities can corrupt the shared representation, leading to a significant drop in performance [36, 51]. In contrast, late fusion treats each modality independently and fuses only the final predictions. As a result, if one modality is missing, its output can be simply ignored, and if one modality is noisy, it does not directly affect the others. This modular structure makes late fusion inherently more robust to missing or noisy modalities.

- **Interpretability.** Latent fusion occurs at the intermediate stages of the model, where fused features are often highly abstract, making it difficult to clearly understand the specific contribution of each modality to the final decision. In contrast, late fusion combines individual modality predictions, allowing inspection of each modality's output separately. This enables easier identification of which modality's prediction caused an error, thereby offering superior interpretability.

We have further added comparisons with latent fusion methods on the MVSA dataset. The experiments are conducted under varying levels of Gaussian noise, and all the competing methods share the same backbone. The compared methods are described as follows:

- **Deep CCA:** Following [52], we implement the CCA loss and combine it with a cross-entropy loss, which performs learning using the shared latent representation.

- **LFM [22]:** LFM integrates contrastive learning into multimodal learning to align features across modalities.

- **SUFA [53]:** SUFA aligns different modalities by minimizing the KL divergence between them. In addition, it constructs positive and negative sample pairs within each unimodal branch for contrastive learning.

Table 7: Compared with latent fusion

| Method | $\epsilon = 0$ | $\epsilon = 5$ | $\epsilon = 10$ |
|--------|------|------|------|
| Deep CCA | 76.49 | 63.58 | 53.94 |
| LFM | 77.26 | 66.47 | 57.80 |
| SUFA | 78.18 | 71.40 | 59.11 |
| MNL | **79.50** | **74.03** | **63.01** |

# C  More Details

## C.1  Image-text Classification

For the image-text classification task, we conduct experiments on the MVSA and UMCP-FOOD101 datasets, following the experimental settings described in [42, 33]. We use a ResNet architecture [54] pretrained on ImageNet [55] as the backbone for the image modality, and a pretrained BERT model [56] for the text modality. The results of competing methods are reported based on the implementations and settings from [42, 33]. All models are trained for 100 epochs with a batch size of 16, using the Adam optimizer. The learning rate is set to 1e-4, with a warmup proportion of 0.1.

## C.2  Senses Recognition

For the sense recognition task on the NYU DEPTH V2 dataset, we compare the proposed method with several multimodal fusion approaches, following the experimental setting [42, 33]. We adopt the ResNet architecture [54], pretrained on ImageNet [55], as the backbone network for each modality. All models are trained for 100 epochs with a batch size of 32, using the Adam optimizer. The learning rate is set to 1e-4, with a warmup proportion of 0.1.

## C.3  Emotion Recognition

For the emotion recognition task, we conduct experiments on the CREMA-D dataset, implementing the baseline and comparison methods based on the settings in [57]. All models are reproduced under the same experimental conditions. We use ResNet-18 [54] as the backbone network, and all models are trained from scratch for the text and visual modalities. Optimizer settings, learning rates, and other hyperparameters follow those used in the respective baseline methods. Specifically, for LATE FUSION, QMF [42], PDF [33], and the variants incorporating MNL, we use the SGD optimizer with a batch size of 64, a learning rate of 0.002, and train for 100 epochs.

## C.4  Multi-LLM Fusion

Recent MLLMs and LLMs [58] primarily rely on feature-level fusion. To demonstrate the generalizability of our approach, we extend MNL to large language models, thereby constructing a multi-LLM fusion setting. Specifically, for both QA and VQA tasks, each individual model, given a prompt, a question, and the corresponding answer candidates, is required to predict only a single token representing the symbol of the chosen option. We collect the logits produced by each model (all models share the same vocabulary) and fuse them with fixed weights at the decision level. We refer to this procedure as LATE FUSION, which allows MNL to be naturally adapted to the multi-LLM fusion setting. A similar strategy can also be applied to MLLMs.

For the QA task, we use the MathQA dataset [45], which contains math word problems annotated with gold answers and step-by-step solution programs. The problems span diverse domains such as arithmetic, algebra, geometry, and probability. MathQA is widely employed to evaluate and improve models' mathematical reasoning and problem-solving capabilities.

For the VQA task, we conduct experiments on the Visual7W dataset [59], a widely used benchmark for visual question answering. Each question in Visual7W is annotated with a gold answer and belongs to one of six categories: what, where, when, who, why, and how. This dataset is commonly used to study image understanding and multimodal reasoning. From Visual7W, we sample data 6993 examples for training and 1400 examples for inference. We fine-tune Qwen2-VL-2B-Instruct and

Qwen2.5-VL-3B-Instruct using LoRA, and compare the performance of LATE FUSION and LATE FUSION + MNL to validate the effectiveness of our method. The results are summarized in Table 8.

Table 8: Performance comparison of VQA task.

| Method | Fusion | Qwen2-VL-2B-Instruct | Qwen2.5-VL-3B-Instruct |
|---|---|---|---|
| LATE FUSION | 84.87 | 83.71 | 84.76 |
| LATE FUSION + MNL | 85.21 | 84.57 | 84.82 |

## C.5 Symbols Table

To avoid potential confusion, we provide a table for main symbols in Table 9

Table 9: Summary of Symbols

| Symbol | Description |
|---|---|
| $\mathcal{M}$ | Modality Set |
| $|\mathcal{M}|$ | Number of Modalities |
| $R(x_i)$ | Multimodal robustness radius of the sample $x_i$ |
| $j$ | The most probable class among the non-target classes |
| $f(x_i)$ | Fusion logits output given sample $x_i$ |
| $f(x_i)_k$ | Fusion logits output for class $k$ given sample $x_i$ |
| $f^{(m)}(x_i^{(m)})$ | Logits output from the $m$-th modality given modality-specific input |
| $\sigma(\cdot)$ | Softmax function |
| $\xi_{(m)}^j$ | UCoM for modality $m$ as defined in Eq. (2), where $j$ denotes the most probable class among the non-target classes |
| $\xi_{(m)}$ | UCoM for modality $m$, in its simplified form, denoted by $\xi_{(m)}^j$ without explicitly specifying the competing class $j$ |
| $\xi'_{(m)}$ | UCoM for modality $m$ after perturbation |
| $\tau_{(m)}$ | Lipschitz constant of modality $m$ |
| $w^{(m)}$ | Weight assigned to modality $m$ in late fusion |
| $P^{(m)}$ | The predicted probability of modality $m$ |
| $P_y^{(m)}$ | The predicted probability of the target class from modality $m$ |

## C.6 Pseudocode of MNL

To facilitate better understanding, we provide pseudocode in Algorithm (1).

---

**Algorithm 1** Multimodal Learning with MNL

---

**Input:** Training set $\mathcal{D}_{train} = \{(x_i, y_i)\}_{i=1}^N \subset \mathcal{X} \times \mathcal{Y}$; Iteration number $T$; Warm-up epoch $W$
**Output:** Trained multimodal model parameters $\theta$
1: **for** $t = 1$ to $T$ **do**
2:     Sample a fresh mini-batch $B_t$ from $\mathcal{D}_{train}$
3:     Feed-forward $B_t$ to the model
4:     Obtain modality-specific logits $f^{(m)}$ and the fusion logits $f(x)$ according to Eq. 1
5:     **if** $t > W$ **then**
6:         Compute each modality $\xi_{(m)}$ (Eq. 3) and identify RDM and IMs according to Definition 3.3.
7:         Compute MNL loss (Eq. 7) and the total loss (Eq. 9)
8:     **else**
9:         Compute cross-entropy loss to warm up
10:     **end if**
11:     Update model parameters $\theta$
12: **end for**

---

## D    Additional Results

### D.1    Additional Ablation Study

In the main paper, we conducted ablation studies on both the guidance scope and guidance conditions. To more intuitively demonstrate the effectiveness of MNL, we performed a comprehensive ablation experiment on the aforementioned conditions, as shown in Table 13.

### D.2    Additional Experiments on Hyperparameters

For the hyperparameter $\lambda$ in Eq. 9, which controls the strength of MNL, our setup sums the MNL loss equally with the CE loss. To illustrate this, we report results on the MVSA dataset, where the CE weight is fixed at 1 while the MNL weight is varied from 0.2 to 2.0 under different noise levels as Table 10.

Table 10: Performance under different noise levels with varying MNL weights.

| $\lambda$ | 0.2 | 0.4 | 0.6 | 0.8 | 1.0 | 1.2 | 1.4 | 1.6 | 1.8 | 2.0 |
|---|---|---|---|---|---|---|---|---|---|---|
| Gauss $\epsilon = 0$ | 76.88 | 78.03 | 79.19 | 78.03 | 80.54 | 79.38 | 79.00 | **81.02** | 80.35 | 79.77 |
| Gauss $\epsilon = 5$ | 72.56 | 72.02 | 73.41 | 73.56 | **74.07** | 73.64 | 72.06 | 73.22 | 73.80 | 72.87 |
| Gauss $\epsilon = 10$ | 62.46 | 62.36 | 62.66 | 63.38 | **63.78** | 63.46 | 62.12 | 61.19 | 63.74 | 63.70 |
| Salt $\epsilon = 5$ | 74.95 | 74.57 | 74.37 | 74.95 | **75.76** | 75.14 | 75.72 | 74.76 | 74.37 | 75.26 |
| Salt $\epsilon = 10$ | 62.27 | 63.09 | 63.46 | 63.27 | 64.93 | **65.06** | 64.08 | 63.50 | 61.96 | 63.01 |

Table 11: Effect of warm-up epochs on accuracy.

| **Warm-up Epoch** | 0 | 2 | 5 | 10 | 15 | 20 | 30 |
|---|---|---|---|---|---|---|---|
| **Accuracy (%)** | 68.55 | 68.11 | 68.11 | 69.18 | 68.79 | 68.26 | 69.05 |

To further investigate the effect of warm-up epoch settings, we report the results on the CREMA-D dataset with noise level 0 in Table 11. We observe that the hyperparameter warm-up epoch is not sensitive (we set it to 10). This may be because the benefit of MNL is relatively small during the early stages of training and becomes more pronounced as training progresses and the modality gap widens, allowing for more meaningful guidance. Therefore, the warm-up is primarily introduced to reduce unnecessary computational costs during the initial training phase.

### D.3    Additional Overhead During Training

Integrating MNL with baseline methods incurs minimal computational overhead and requires no modification to the underlying network architecture. During training, MNL operates solely at the output level, identifying the Robust Dominant Modality (RDM) and the Inferior Modality (IM), determining the guidance direction, and computing the additional loss. At inference time, the procedure is identical to that of the baseline, introducing no extra cost. Specifically, we calculate the average time per training iteration (per batch) on MVSA, FOOD101, NYUDv2, and CREMAD, respectively. As shown in Table 12, the additional overhead introduced by MNL during training is negligible.

Table 12: Inference time (in milliseconds) of different methods on various datasets.

| **Method** | **MVSA** | **FOOD101** | **NYUDv2** | **CREMA-D** |
|---|---|---|---|---|
| LATE FUSION | 20.96 ms | 69.23 ms | 32.27 ms | 63.03 ms |
| LATE FUSION + MNL | 31.15 ms | 85.76 ms | 49.19 ms | 87.11 ms |

### D.4 Full Results with Standard Deviation

In this section, we present the full results with standard deviation in Table 14 and Table 15.

Table 13: Performance comparison under different guidance scopes and conditions across various Gaussian perturbation levels on the MVSA dataset, where red and blue indicates the best/runner-up performance.

| All-Class | Non-target | Prior | Confident | Robust | $\epsilon = 0$ | $\epsilon = 5$ | $\epsilon = 10$ |
|:---:|:---:|:---:|:---:|:---:|:---:|:---:|:---:|
| ✓ | | ✓ | | | $78.77 \pm 1.52$ | $72.77 \pm 0.36$ | $62.36 \pm 2.09$ |
| ✓ | | | ✓ | | $78.47 \pm 0.87$ | $72.74 \pm 0.62$ | $62.62 \pm 1.40$ |
| ✓ | | | ✓ | ✓ | $78.90 \pm 0.29$ | $72.16 \pm 0.48$ | $62.52 \pm 3.95$ |
| | ✓ | ✓ | | | $78.66 \pm 0.87$ | $72.69 \pm 1.07$ | $62.77 \pm 1.62$ |
| | ✓ | | ✓ | | $78.74 \pm 0.79$ | $71.87 \pm 1.93$ | $59.35 \pm 0.16$ |
| | ✓ | | ✓ | ✓ | $79.50 \pm 1.70$ | $74.03 \pm 1.11$ | $63.01 \pm 1.74$ |

## E Limitations and Broader Impacts

The limitations of MNL can be better understood by examining the performance gap between modalities across datasets, as shown in Fig. 8. In the CREMA-D dataset, the audio modality outperforms the visual modality by a large margin, whereas in the NYU Depth V2 dataset, the RGB modality slightly outperforms the depth modality. The modality gap in CREMA-D is noticeably larger than in NYU Depth V2. Since MNL primarily relies on leveraging the stronger modality to guide the weaker one, its benefits may be limited when the performance gap is small and both modalities improve in sync. These observations suggest that the effectiveness of MNL is closely tied to the degree of modality complementarity and asymmetry present in the data.

Reviewing Equation 20, the lower bound of multimodal robustness is determined by both the modality weights and the UCoM. MNL aims to improve the robustness of the multimodal system by enhancing the UCoM of weaker modalities. In contrast, dynamic late fusion methods typically assign higher weights to modalities that are more confident in predicting the ground truth class. This fundamental difference introduces a degree of incompatibility between MNL and existing dynamic weight allocation strategies. As a result, the performance gains achieved by MNL are often smaller than those observed with static late fusion methods. In future work, it would be valuable to explore whether dynamic weight allocation can be guided by UCoM, so that the enhancements MNL provides to weaker modalities are not undermined by their lower assigned weights.

Furthermore, MNL can be directly integrated with most late fusion methods, with the exception of certain approaches such as evidence-based late fusion. In contrast, how to effectively incorporate MNL into early fusion frameworks remains an open question, which we leave for future work.

Regarding the potential social impact, this paper proposes a method that leverages robust dominant modalities to guide inferior modalities in suppressing non-target classes, effectively enhancing the robustness of multimodal systems under noisy conditions. This approach holds promise for safety-critical applications such as human-computer interaction, helping multimodal systems remain stable and reliable in complex environments. However, due to the high uncertainty and uncontrollability of noise in open environments, certain risks may still exist in high-stakes scenarios like autonomous driving and medical diagnosis.

Table 14: Performance of different methods under varying noise levels across four (MVSA, FOOD101, NYU Depth V2, and CREMA-D) datasets. We add Gaussian noise on 50% modalities and $\varepsilon$ presents the noise degree.

| DATASET | METHOD | $\epsilon = 0.0$ | $\epsilon = 5.0$ | $\epsilon = 10.0$ |
|---|---|---|---|---|
| MVSA | IMG | $64.12 \pm 1.23$ | $49.36 \pm 2.02$ | $45.00 \pm 2.63$ |
| | TEXT | $75.61 \pm 0.53$ | $69.50 \pm 1.50$ | $47.41 \pm 0.79$ |
| | LATE FUSION | $76.88 \pm 1.30$ | $63.46 \pm 3.46$ | $55.16 \pm 3.60$ |
| | DYNMM | $79.07 \pm 0.53$ | $67.96 \pm 1.65$ | $59.21 \pm 1.41$ |
| | TMC | $74.87 \pm 2.24$ | $66.72 \pm 4.55$ | $60.35 \pm 2.79$ |
| | QMF | $78.07 \pm 1.10$ | $73.85 \pm 1.42$ | $61.28 \pm 2.12$ |
| | PDF | $79.94 \pm 0.95$ | $74.40 \pm 1.51$ | $63.09 \pm 1.33$ |
| | LATE FUSION+MNL | $79.50 \pm 1.70$ | $74.03 \pm 1.11$ | $63.01 \pm 1.74$ |
| | QMF+MNL | $79.45 \pm 0.60$ | $74.12 \pm 0.06$ | $62.75 \pm 0.04$ |
| | PDF+MNL | $80.54 \pm 0.85$ | $74.07 \pm 1.93$ | $63.78 \pm 1.27$ |
| UMPC FOOD 101 | IMG | $64.62 \pm 0.40$ | $34.72 \pm 0.53$ | $33.03 \pm 0.37$ |
| | TEXT | $86.46 \pm 0.05$ | $67.38 \pm 0.19$ | $43.88 \pm 0.32$ |
| | LATE FUSION | $90.69 \pm 0.12$ | $68.49 \pm 3.37$ | $57.99 \pm 1.59$ |
| | DYNMM | $92.59 \pm 0.07$ | $74.74 \pm 0.19$ | $59.68 \pm 0.20$ |
| | TMC | $89.86 \pm 0.07$ | $73.93 \pm 0.34$ | $61.37 \pm 0.21$ |
| | QMF | $92.90 \pm 0.11$ | $76.03 \pm 0.70$ | $62.21 \pm 0.25$ |
| | PDF | $93.32 \pm 0.22$ | $76.47 \pm 0.31$ | $62.83 \pm 0.31$ |
| | LATE FUSION+MNL | $92.77 \pm 0.17$ | $75.16 \pm 0.80$ | $62.06 \pm 0.49$ |
| | QMF+MNL | $93.03 \pm 0.16$ | $75.41 \pm 0.48$ | $62.59 \pm 0.55$ |
| | PDF+MNL | $93.33 \pm 0.22$ | $76.65 \pm 0.31$ | $63.16 \pm 0.32$ |
| NYU DEPTH V2 | DEPTH | $63.30 \pm 0.48$ | $53.12 \pm 1.52$ | $45.46 \pm 2.07$ |
| | RGB | $62.65 \pm 1.22$ | $50.95 \pm 3.38$ | $44.13 \pm 3.80$ |
| | LATE FUSION | $70.03 \pm 0.84$ | $64.37 \pm 0.80$ | $60.55 \pm 1.65$ |
| | DYNMM | $65.50 \pm 0.37$ | $54.31 \pm 1.72$ | $46.79 \pm 1.09$ |
| | TMC | $70.40 \pm 0.31$ | $59.33 \pm 2.19$ | $50.61 \pm 2.87$ |
| | QMF | $69.54 \pm 1.06$ | $64.10 \pm 1.42$ | $60.18 \pm 1.23$ |
| | PDF | $71.37 \pm 0.76$ | $65.72 \pm 1.72$ | $62.56 \pm 1.84$ |
| | LATE FUSION+MNL | $71.05 \pm 1.06$ | $67.02 \pm 0.56$ | $63.81 \pm 1.63$ |
| | QMF+MNL | $71.25 \pm 0.59$ | $65.38 \pm 1.09$ | $61.80 \pm 1.52$ |
| | PDF+MNL | $71.52 \pm 0.89$ | $67.01 \pm 1.72$ | $63.07 \pm 0.50$ |
| CREMA-D | AUDIO | $60.70 \pm 0.94$ | $59.60 \pm 1.40$ | $49.52 \pm 2.91$ |
| | VISUAL | $56.23 \pm 1.67$ | $52.47 \pm 2.06$ | $38.17 \pm 3.74$ |
| | LATE FUSION | $68.04 \pm 3.92$ | $64.25 \pm 6.68$ | $52.39 \pm 7.40$ |
| | DYNMM | $63.27 \pm 0.68$ | $62.01 \pm 1.27$ | $51.43 \pm 1.24$ |
| | TMC | $63.63 \pm 1.16$ | $62.68 \pm 1.17$ | $57.97 \pm 2.14$ |
| | QMF | $66.13 \pm 1.32$ | $64.27 \pm 1.47$ | $50.77 \pm 5.83$ |
| | PDF | $67.07 \pm 0.83$ | $64.57 \pm 1.94$ | $53.33 \pm 2.78$ |
| | LATE FUSION+MNL | $73.71 \pm 2.10$ | $70.35 \pm 1.27$ | $57.26 \pm 4.28$ |
| | QMF+MNL | $68.18 \pm 0.35$ | $67.00 \pm 0.95$ | $52.62 \pm 2.96$ |
| | PDF+MNL | $69.18 \pm 0.96$ | $66.94 \pm 1.82$ | $55.43 \pm 4.61$ |

Table 15: Performance of different methods under varying noise levels across four (MVSA, FOOD101, NYU Depth V2, and CREMA-D) datasets. We add Salt noise on 50% modalities and $\varepsilon$ presents the noise degree.

| DATASET | METHOD | $\epsilon = 0.0$ | $\epsilon = 5.0$ | $\epsilon = 10.0$ |
|---|---|---|---|---|
| MVSA | IMG | $64.12 \pm 1.23$ | $56.72 \pm 1.92$ | $50.71 \pm 3.20$ |
| | TEXT | $75.61 \pm 0.53$ | $69.50 \pm 1.50$ | $47.41 \pm 0.79$ |
| | LATE FUSION | $76.88 \pm 1.30$ | $67.88 \pm 1.87$ | $55.43 \pm 1.94$ |
| | DYNMM | $79.07 \pm 0.53$ | $71.35 \pm 0.97$ | $59.96 \pm 1.31$ |
| | TMC | $74.87 \pm 2.24$ | $68.02 \pm 3.07$ | $56.62 \pm 3.67$ |
| | QMF | $78.07 \pm 1.10$ | $73.90 \pm 1.89$ | $60.41 \pm 2.63$ |
| | PDF | $79.94 \pm 0.95$ | $75.11 \pm 1.15$ | $61.97 \pm 1.14$ |
| | LATE FUSION+MNL | $79.50 \pm 1.70$ | $74.68 \pm 1.43$ | $62.31 \pm 1.71$ |
| | QMF+MNL | $79.45 \pm 0.60$ | $75.14 \pm 1.19$ | $64.68 \pm 1.94$ |
| | PDF+MNL | $80.54 \pm 0.85$ | $75.76 \pm 0.48$ | $64.93 \pm 1.37$ |
| UMPC FOOD 101 | IMG | $64.62 \pm 0.40$ | $50.75 \pm 0.44$ | $36.83 \pm 0.92$ |
| | TEXT | $86.46 \pm 0.05$ | $67.38 \pm 0.19$ | $43.88 \pm 0.32$ |
| | LATE FUSION | $90.69 \pm 0.16$ | $77.99 \pm 0.54$ | $58.75 \pm 0.99$ |
| | DYNMM | $92.59 \pm 0.07$ | $78.91 \pm 0.20$ | $57.64 \pm 0.30$ |
| | TMC | $89.86 \pm 0.07$ | $77.86 \pm 0.41$ | $60.22 \pm 0.43$ |
| | QMF | $92.90 \pm 0.13$ | $80.87 \pm 0.40$ | $61.60 \pm 0.20$ |
| | PDF | $93.32 \pm 0.22$ | $81.21 \pm 0.34$ | $61.76 \pm 0.33$ |
| | LATE FUSION+MNL | $92.77 \pm 0.17$ | $80.87 \pm 0.57$ | $61.41 \pm 1.49$ |
| | QMF+MNL | $93.03 \pm 0.16$ | $81.14 \pm 0.52$ | $62.47 \pm 0.38$ |
| | PDF+MNL | $93.33 \pm 0.22$ | $81.52 \pm 0.27$ | $62.95 \pm 0.19$ |
| NYU DEPTH V2 | DEPTH | $63.30 \pm 0.48$ | $50.99 \pm 1.41$ | $38.56 \pm 2.16$ |
| | RGB | $62.65 \pm 1.22$ | $49.14 \pm 1.40$ | $34.76 \pm 1.59$ |
| | LATE FUSION | $70.03 \pm 0.84$ | $62.05 \pm 1.17$ | $51.50 \pm 1.81$ |
| | DYNMM | $65.50 \pm 0.37$ | $52.26 \pm 1.45$ | $38.17 \pm 1.17$ |
| | TMC | $70.40 \pm 0.31$ | $59.33 \pm 1.47$ | $45.32 \pm 2.84$ |
| | QMF | $69.54 \pm 1.06$ | $62.02 \pm 1.47$ | $51.87 \pm 0.91$ |
| | PDF | $71.37 \pm 0.76$ | $64.27 \pm 1.36$ | $53.62 \pm 2.15$ |
| | LATE FUSION+MNL | $71.05 \pm 1.06$ | $64.88 \pm 0.97$ | $54.08 \pm 2.34$ |
| | QMF+MNL | $71.25 \pm 0.59$ | $63.62 \pm 1.62$ | $53.30 \pm 2.60$ |
| | PDF+MNL | $71.52 \pm 0.89$ | $64.32 \pm 0.58$ | $54.08 \pm 0.83$ |
| CREMA-D | AUDIO | $60.70 \pm 0.94$ | $59.60 \pm 1.40$ | $49.52 \pm 2.91$ |
| | VISUAL | $56.23 \pm 1.67$ | $50.62 \pm 3.27$ | $43.90 \pm 2.80$ |
| | LATE FUSION | $68.04 \pm 3.92$ | $62.50 \pm 6.63$ | $52.61 \pm 6.51$ |
| | DYNMM | $63.27 \pm 1.74$ | $62.92 \pm 0.98$ | $52.33 \pm 3.57$ |
| | TMC | $63.63 \pm 1.16$ | $62.31 \pm 1.93$ | $58.44 \pm 3.16$ |
| | QMF | $66.13 \pm 1.32$ | $63.73 \pm 1.11$ | $51.55 \pm 5.31$ |
| | PDF | $67.07 \pm 0.83$ | $63.44 \pm 2.39$ | $53.71 \pm 2.40$ |
| | LATE FUSION+MNL | $73.71 \pm 2.10$ | $68.06 \pm 1.25$ | $58.41 \pm 4.70$ |
| | QMF+MNL | $68.18 \pm 0.35$ | $64.05 \pm 2.93$ | $52.32 \pm 2.67$ |
| | PDF+MNL | $69.18 \pm 0.96$ | $63.77 \pm 2.24$ | $54.30 \pm 4.84$ |

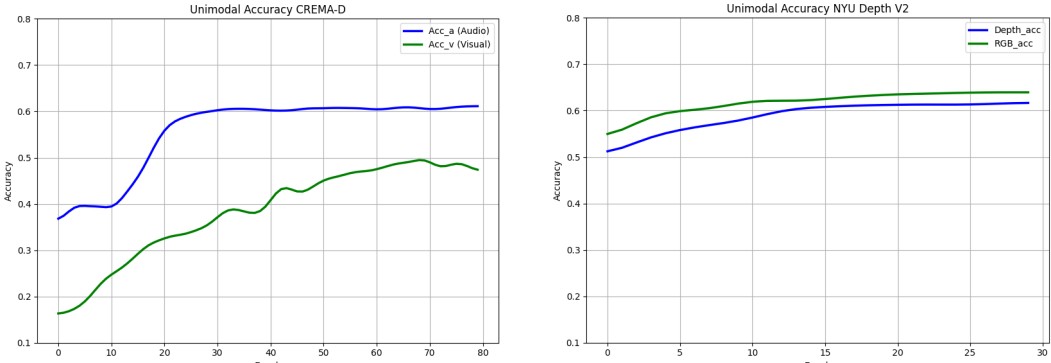

Figure 8: Unimodal performance on the CREMA-D and NYU Depth V2 datasets

