# OpenReview forum: "Multimodal Negative Learning"
_NeurIPS.cc/2025/Conference — NeurIPS 2025 poster_

### Official Review · Reviewer_v4mL · 2025-06-30

**Clarity:** 4
**Significance:** 4
**Originality:** 4
**Rating:** 5
**Confidence:** 5

**Summary:**

Considering the multimodal imbalance of “Learning to be” in existing methods, this work proposes a new “Learning Not to be” paradigm, which leverages the dominant modality to dynamically guide the weak modality on non-target classes. The authors provided an interesting multimodal negative learning framework, which theoretically tightens the robustness lower bound of multimodal learning by increasing the Unimodal Confidence Margin (UCoM). In addition, the proposed MNL can be flexibly integrated with late fusion methods, such as QMF and PDF, which has the potential to be applied in broader scenarios. The comparisons on noisy data and ablated experiments have validated its effectiveness. Generally speaking, this work contributes new insights to the multimodal learning community.

**Questions:**

1. What’s the key effect of dynamic guidance on multimodal negative learning? Eq. 6 should be further explained.
2. Is there any trade-off between multimodal positive learning and multimodal negative learning in Eq. 7?
3. There are two stages during the training phase. How to determine stages 1 and 2 in practice? More details should be provided.
4. Although the proposed method can be integrated with existing methods without introducing additional inference overhead, the additional training cost should be reported.

**Ethical Concerns:**

["NO or VERY MINOR ethics concerns only"]

**Final Justification:**

After considering the authors' rebuttal and the other reviewers' comments, I maintain my overall rating of 5.

**Limitations:**

yes

**Quality:**

4

**Strengths And Weaknesses:**

### Strengths:
1. The proposed multimodal negative learning framework is novel and interesting, which changes the commonly used “Learning to be” to “Learning Not to be” paradigm. It allows weak modalities to preserve unique information without being over-aligned, stabilizing the decision space and preserving the specificity of individual modality.
2. This work introduces a dynamic guidance mechanism from the robustness perspective. The proposed UCoM provably tightens the robustness lower bound of multimodal learning and reduces the empirical error of weak modalities.
3. This work is verified on multiple benchmarks and has conducted experiments on extensive noisy and imbalanced scenarios, demonstrating its effectiveness and generalization.
4. This work can be flexibly integrated with many existing methods without additional inference consumption and consistently improve their performance on multiple datasets.
5. This work provides theoretical proof on robust lower bounds and empirical error in Supp. A, offering theoretical guarantees to mitigate modality imbalance and boost robustness.

### Weaknesses:
1. The authors conducted extensive experiments, however, the competing method “LATE FUSION” is confusing for me. QMF and PDF also belong to late fusion methods. Therefore, the claimed “LATE FUSION” is not inappropriate.
2. The authors provide theoretical proof in supplementary material, while some key equations should be presented in the main paper to make it easier to follow.
3. The presentation can be further improved, especially for the tables and figures. For instance, the font size of Tables 3 and 4 is too large, while the font is too small in Fig. 3.
4. It is reasonable to consider the dynamic modality dominance in multimodal learning, while a more clear illustration will be helpful.
5. Since this work can be flexibly integrated with existing methods, I suggest the authors present the pseudo code in supplementary material.

---

> ### Author Rebuttal · Authors · 2025-07-31
>
> We sincerely thank the reviewer for the valuable comments, the recognition of our method’s novelty and theoretical contributions, and the acknowledgment of its potential benefits to broader multimodal learning research. We believe the constructive feedback will help us further improve the paper and enhance its impact on the community.
>
> > W1 The authors conducted extensive experiments, however, the competing method “LATE FUSION” is confusing for me. QMF and PDF also belong to late fusion methods. Therefore, the claimed “LATE FUSION” is not inappropriate.
>
> Thanks for your comment. The "LATE FUSION" mentioned here specifically refers to "STATIC LATE FUSION." We have corrected this in the manuscript and will ensure the final version reflects the change.
>
> > W4 & Q1 It is reasonable to consider the dynamic modality dominance in multimodal learning, while a more clear illustration will be helpful. & What’s the key effect of dynamic guidance on multimodal negative learning? Eq. 6 should be further explained.
>
> Thanks for your question. The dynamic guidance mechanism is indeed a critical component of MNL. Its core is defined in Definition 3.3, which jointly considers both $P _y$ and the Unimodal Confidence Margin (UCoM) to identify the Robust Dominant Modality (RDM) and the Inferior Modalities (IMs). Specifically, **the RDM is selected such that its $P _y$ and UCoM are both greater than those of the IMs, thereby enabling the guidance direction to be determined dynamically.**
>
> Regarding Eq. (6), its key function is to determine two aspects: the guidance direction, which involves identifying the RDM and IMs (as described in Definition 3.3, with Eq. (6) focusing on the two-modality case), and the guidance scope, which is restricted to non-target classes.
> **If dynamic guidance is absent in multimodal negative learning:**
>
> - A modality that predicts incorrectly might guide a modality that predicts correctly. Even if the guidance is restricted to non-target classes, this can still disrupt the optimization process of the modalities.
> - A modality with a low Unimodal Confidence Margin (UCoM) might guide a modality with a high UCoM, which can potentially harm the robustness lower bound of the multimodal system.
>
> > W5 Since this work can be flexibly integrated with existing methods, I suggest the authors present the pseudo code in supplementary material.
>
> Thank you for the excellent suggestion. Below, we provide the pseudocode of the algorithm. The inclusion of this pseudocode will be reflected in the final version of the paper.
>
> ```text
> Algorithm 1: Multimodal Learning with MNL
>
> Input:
>  Training set D_train = {(x_i, y_i)}_{i=1}^N ⊂ X × Y
>  Iteration number T, warm-up epoch W
>
> for t = 1 to T do
>     Sample a fresh mini-batch B_t from D_train
>     Feed-forward the batched data B_t to the model
>     Obtain fusion logits and modality-specific logits from the model
>
>     if t > W then
>         Identify RDM and IMs according to Definition 3.3
>         Compute MNL loss according to Eq. (5)
>         Compute total loss according to Eq. (7)
>     else
>         Compute cross-entropy loss
>     end if
>
>     Update model parameters
> end for
> ```
>
> > Q2 Is there any trade-off between multimodal positive learning and multimodal negative learning in Eq. 7?
>
> Thanks for your question. We present our results on the MVSA dataset, where the CE weight is fixed at 1 and the MNL weight is varied from 0.2 to 2.0 under different noise levels.
>
>
> | weight   | 0.2    | 0.4    | 0.6    | 0.8    | 1.0        | 1.2        | 1.4    | 1.6        | 1.8    | 2.0    |
> | -------- | ------ | ------ | ------ | ------ | ---------- | ---------- | ------ | ---------- | ------ | ------ |
> | Gauss_0  | 0.7688 | 0.7803 | 0.7919 | 0.7803 | 0.8054     | 0.7938     | 0.7900 | **0.8102** | 0.8035 | 0.7977 |
> | Gauss_5  | 0.7256 | 0.7202 | 0.7341 | 0.7356 | **0.7407** | 0.7364     | 0.7206 | 0.7322     | 0.7380 | 0.7287 |
> | Gauss_10 | 0.6246 | 0.6236 | 0.6266 | 0.6338 | **0.6378** | 0.6346     | 0.6212 | 0.6119     | 0.6374 | 0.6370 |
> | Salt_5   | 0.7495 | 0.7457 | 0.7437 | 0.7495 | **0.7576** | 0.7514     | 0.7572 | 0.7476     | 0.7437 | 0.7526 |
> | Salt_10  | 0.6227 | 0.6309 | 0.6346 | 0.6327 | 0.6493     | **0.6506** | 0.6408 | 0.6350     | 0.6196 | 0.6301 |
>
> Similar results are observed across multiple datasets. After careful consideration, we set the weights of both losses (CE and MNL) to be equal at 1.
>
> > Q3 There are two stages during the training phase. How to determine stages 1 and 2 in practice? More details should be provided.
>
> Thank you for your question. We can provide the table (CREMA-D dataset, Noise = 0).
>
>
> | Warm-up Epoch | 0     | 2     | 5     | 10    | 15    | 20    | 30    |
> | ------------- | ----- | ----- | ----- | ----- | ----- | ----- | ----- |
> | Accuracy (%)  | 68.55 | 68.11 | 68.11 | 69.18 | 68.79 | 68.26 | 69.05 |
>
> We observe that the hyperparameter warm-up epoch is not sensitive (we set it to 10). This is because the benefit of MNL is relatively small in the early stages of training, and becomes more effective as training progresses and the modality gap becomes more pronounced, allowing for more meaningful guidance. Therefore, the warm-up is primarily introduced to reduce unnecessary computational cost during the initial training phase.
>
> > Q4 Although the proposed method can be integrated with existing methods without introducing additional inference overhead, the additional training cost should be reported.
>
> We are glad to report the additional training cost. Specifically, we calculate the average time per training iteration (per batch), with batch sizes of 16, 16, 32, and 64 for MVSA, FOOD101, NYUDv2, and CREMAD, respectively.
>
>
> | Method                   | MVSA    | FOOD101 | NYUDv2  | CREMAD  |
> | ------------------------ | ------- | ------- | ------- | ------- |
> | STATIC LATE FUSION       | 20.96ms | 69.23ms | 32.27ms | 63.03ms |
> | STATIC LATE FUSION + MNL | 31.15ms | 85.76ms | 49.19ms | 87.11ms |
>
> These results indicate that the additional overhead introduced by MNL is minimal.
>
> >presentation issues
>
> Thank you for pointing out the typo. We have revised the description of “LATE FUSION,” improved the clarity of the figures and tables, and made the equations more comprehensible. Finally, we have carefully reviewed and corrected all representation issues throughout the paper.

---

> > ### Comment · Reviewer_v4mL · 2025-08-04
> > **Thanks for responding**
> >
> > I appreciate the authors' comprehensive response, which has effectively addressed my concerns. This method is well-motivated to perform “learning not to be” in multi-modal learning. As few works explore dynamic negative guidance from a robustness perspective, the proposed multimodal negative learning is remarkably novel.
> >
> > I have carefully reviewed the authors’ rebuttals to other reviewers, and I find the further clarifications, particularly the extended evaluations involving LLMs, empirically demonstrate its generalizability and effectiveness. Consequently, I tend to maintain my overall rating of 5: Accept.

---

### Official Review · Reviewer_d8o7 · 2025-07-01

**Clarity:** 3
**Significance:** 3
**Originality:** 2
**Rating:** 4
**Confidence:** 3

**Summary:**

This paper introduces a new paradigm for multimodal learning called Multimodal Negative Learning (MNL). The core idea is to address the "modality imbalance" problem, where a dominant modality can overwhelm a weaker one during training. Instead of traditional "Positive Learning" that forces the weak modality to mimic the strong one's predictions for the correct class, MNL uses the dominant modality to guide the weak modality to suppress non-target classes.

**Questions:**

1. What are the loss weights in Eq. 7 (is MNL summed equally with CE losses?)
2. What are the criteria for transitioning Stage 1→Stage 2 training?
3. Fig 1(b) shows forced alignment harms weak modalities. Does MNL mitigate such collapse? It will be beneficial to provide examples where MNL fails to preserve weak-modality advantages, and quantify how often this occurs.

**Ethical Concerns:**

["NO or VERY MINOR ethics concerns only"]

**Final Justification:**

most of the questions have been addressed. However after the experiments of MNL (step1) + PDF (step2), the performance gain is still small when comparing with late fusion in table 1. Therefore, I'd like to maintain my score.

**Limitations:**

Yes

**Quality:**

3

**Strengths And Weaknesses:**

**Strength**:

1. The paper addresses a well-known challenge in multimodal learning: modality imbalance and over-alignment issues.

2. The paper attempts to theoretically justify its approach (lower bounds based on UCoM) by linking it to the robustness of the multimodal system.

3. Experiments are conducted on four publicly available datasets with different modalities (vision, text, audio), providing a broad evaluation.

4. Thorough ablation studies on different design choices.

**Weakness**:

1. In table 1 and table 2, the improvement from the baseline is not consistently strong. For instance, when MNL is applied to state-of-the-art dynamic fusion methods like PDF and QMF, the performance gains are often marginal, sometimes less than 1%.

2. Theorem 3.1 assumes a statis fusion weights, yet experiments include dynamic fusion, creating a gap between theory and empirical results.

3.  Equation 5 assembles knowledge distillation, where the "soft labels" from a teacher (RDM) are used to train a student (IM), diminishing the novelty of the paper.

---

> ### Author Rebuttal · Authors · 2025-07-31
>
> We'd like to thank the reviewer for the valuable comments and the recognition of our novel “Learn Not to Be” paradigm. We appreciate your support and constructive suggestions, and we address your concerns in detail as follows.
>
> > W1 In table 1 and table 2, the improvement from the baseline is not consistently strong. For instance, when MNL is applied to state-of-the-art dynamic fusion methods like PDF and QMF, the performance gains are often marginal, sometimes less than 1%.
>
> Thank you for your valuable comment. Taking PDF [1] as an example, PDF aims to effectively leverage multimodal data, whereas **MNL focuses on enhancing the weak model's perceptual ability**. Therefore, when we follow the standard setting and directly integrate the two methods, their respective advantages may not be fully exploited. For dynamic fusion methods such as PDF, we suggest a two-step training procedure: First, use MNL to enhance the model’s recognition ability, especially for the weaker modality (Step 1), and then apply PDF to dynamically fuse informative content from different modalities (Step 2).
>
> Specifically, during training:
>
> - **Step 1**: We perform multimodal fusion based on static late fusion and apply the MNL, while simultaneously training PDF to learn the ability to predict fusion weights $ w $.
> - **Step 2**: Based on the model obtained in Step 1, we utilize PDF to predict $ w $ and conduct dynamic fusion accordingly.
>
> To validate this setting, we have conducted additional experiments on the MVSA dataset under Gaussian noise with varying levels of noise (ε = 0, ε = 5, ε = 10). The results are summarized in the following table (over 5 seeds).
>
>
> | Method                        | ε = 0    | ε = 5    | ε = 10   |
> | ----------------------------- | --------- | --------- | --------- |
> | PDF                           | 79.94     | 74.40     | 63.09     |
> | PDF + MNL                     | 80.54     | 74.07     | 63.78     |
> | **MNL (step1) + PDF (step2)** | **80.60** | **74.82** | **63.85** |
>
> The most notable improvement occurs when ε = 5, where the performance of MNL (step1) + PDF (step2) surpasses that of PDF by 0.42%, in contrast to the 0.33% drop observed with PDF + MNL. The MNL (step1) + PDF (step2) configuration not only enhances the performance of the weaker modality but also dynamically allocates weights at inference based on data quality, thereby delivering a more effective boost to overall model performance.
>
> > W2 Theorem 3.1 assumes a statis fusion weights, yet experiments include dynamic fusion, creating a gap between theory and empirical results.
>
> Thanks for your comment. **MNL supports dynamic fusion**, but according to Eq. (4), the dynamic weight $w$ is expected to be positively correlated with $\xi$. However, since $\xi$ is difficult to predict in practice, it becomes challenging to adjust the weights accordingly in a truly dynamic manner. Therefore, we adopt a static fusion assumption.
>
> As indicated by Eq. (4), when $w$ is fixed, the robustness lower bound improves more steadily; when $w$ is dynamic, the robustness gain varies depending on how the weights are set, which may influence the overall performance improvement. This also explains why the benefit of combining our method with existing dynamic fusion approaches such as PDF is somewhat limited.
>
> To mitigate the influence of dynamic weighting on MNL, we first training a static multimodal fusion system using MNL to enhance the model’s ability to perceive weak modalities, and then applying a dynamic fusion strategy like PDF to further boost overall performance (**MNL (step1) + PDF (step2)**).
>
> > W3 Equation 5 assembles knowledge distillation, where the "soft labels" from a teacher (RDM) are used to train a student (IM), diminishing the novelty of the paper.
>
> Thanks for your valuable comment. We propose a new MNL learning paradigm inspired by knowledge distillation (KD).
>
> Differently, KD focuses on training a student model under the guidance of a fixed teacher model, whereas our method focuses on dynamically enhancing the model’s ability to recognize non-target classes of the weak modality by leveraging the dominant modality.
>
> In addition, beyond the commonly used KL divergence in KD, we also explore using L1 and L2 losses to implement MNL. We believe that our MNL has the potential to be compatible with more collaborative learning paradigms and adaptable to a wider range of tasks.
>
> To explore this, we extended our experiments by modifying the MNL loss formulation (Eq. (5)) and implemented negative learning using  L1 or L2 losses.
> Table 1 MVSA dataset under Gaussian noise (over 5 seeds)
>
>
> | Method                       | ε = 0 | ε = 5 | ε = 10 |
> | ---------------------------- | ------ | ------ | ------- |
> | STATIC LATE FUSION           | 76.88  | 63.46  | 55.16   |
> | STATIC LATE FUSION + MNL     | 79.50  | 74.03  | 63.01   |
> | STATIC LATE FUSION + MNL(L2) | 79.48  | 73.48  | 62.96   |
> | STATIC LATE FUSION + MNL(L1) | 79.90  | 73.15  | 59.73   |
>
> Table 2 MVSA dataset under Salt noise (over 5 seeds)
>
>
> | Method                       | ε = 0 | ε = 5 | ε = 10 |
> | ---------------------------- | ------ | ------ | ------- |
> | STATIC LATE FUSION           | 76.88  | 67.88  | 55.43   |
> | STATIC LATE FUSION + MNL     | 79.50  | 74.68  | 62.31   |
> | STATIC LATE FUSION + MNL(L2) | 79.48  | 74.59  | 63.37   |
> | STATIC LATE FUSION + MNL(L1) | 79.90  | 73.73  | 59.02   |
>
> > Q1 What are the loss weights in Eq. 7 (is MNL summed equally with CE losses?)
>
> Yes, according to our setup, the MNL loss is summed equally with the CE loss. To demonstrate this, we present our results on the MVSA dataset, where the CE weight is fixed at 1 and the MNL weight is varied from 0.2 to 2.0 under different noise levels.
>
>
> | weight   | 0.2    | 0.4    | 0.6    | 0.8    | 1.0        | 1.2        | 1.4    | 1.6        | 1.8    | 2.0    |
> | -------- | ------ | ------ | ------ | ------ | ---------- | ---------- | ------ | ---------- | ------ | ------ |
> | Gauss_0  | 0.7688 | 0.7803 | 0.7919 | 0.7803 | 0.8054     | 0.7938     | 0.7900 | **0.8102** | 0.8035 | 0.7977 |
> | Gauss_5  | 0.7256 | 0.7202 | 0.7341 | 0.7356 | **0.7407** | 0.7364     | 0.7206 | 0.7322     | 0.7380 | 0.7287 |
> | Gauss_10 | 0.6246 | 0.6236 | 0.6266 | 0.6338 | **0.6378** | 0.6346     | 0.6212 | 0.6119     | 0.6374 | 0.6370 |
> | Salt_5   | 0.7495 | 0.7457 | 0.7437 | 0.7495 | **0.7576** | 0.7514     | 0.7572 | 0.7476     | 0.7437 | 0.7526 |
> | Salt_10  | 0.6227 | 0.6309 | 0.6346 | 0.6327 | 0.6493     | **0.6506** | 0.6408 | 0.6350     | 0.6196 | 0.6301 |
>
> Similar results are observed across multiple datasets. After careful consideration, we set the weights of both losses to be equal at 1.
>
> > Q2 What are the criteria for transitioning Stage 1→Stage 2 training?
>
> Thank you for your question. We can provide the following table (CREMA-D dataset, Noise = 0).
>
>
> | Warm-up Epoch | 0     | 2     | 5     | 10    | 15    | 20    | 30    |
> | ------------- | ----- | ----- | ----- | ----- | ----- | ----- | ----- |
> | Accuracy (%)  | 68.55 | 68.11 | 68.11 | 69.18 | 68.79 | 68.26 | 69.05 |
>
> We observe that the hyperparameter warm-up epoch is not sensitive. We set it to 10.
>
> > Q3 Fig 1(b) shows forced alignment harms weak modalities. Does MNL mitigate such collapse? It will be beneficial to provide examples where MNL fails to preserve weak-modality advantages, and quantify how often this occurs.
>
> Thank you for your question. First, regarding whether MNL can mitigate such collapse, the following table presents a comparison of MNL (warm-up epoch = 10) and KL Guidance in terms of the number of Audio-wrong but Video-right samples across different epochs.
>
>
> | Method      | 10  | 11  | 12  | 13  | 14  | 15  | 16  | 17  | 18 | 19 | 20 |
> | ----------- | --- | --- | --- | --- | --- | --- | --- | --- | -- | -- | -- |
> | KL Guidance | 446 | 385 | 324 | 252 | 212 | 137 | 158 | 122 | 77 | 36 | 29 |
> | MNL         | 392 | 300 | 266 | 145 | 128 | 119 | 94  | 28  | 7  | 4  | 2  |
>
> It is evident that MNL consistently results in significantly fewer audio-wrong but video-right samples compared to KL Guidance at the same training epoch.
> - In KL Guidance, the audio modality (which generally performs better) is used to guide the video modality even when its own prediction is incorrect and the video prediction is correct. This misalignment introduces a conflict between the cross-entropy loss and the KL divergence, often leading to optimization collapse.
>
> - In contrast, MNL dynamically adjusts the guidance direction based on Definition 3.3, thereby avoiding such conflicts.
>
> As for cases where MNL fails to preserve weak-modality advantages, we take the MVSA dataset as an example. When applying MNL, the guidance direction is determined dynamically based on $P_y$ and $\xi$(UCoM), and according to Eq. (6), the guiding modality must have both higher $P_y$ and $\xi$ than the guided one. However, this condition is not always satisfied for every sample. Out of **1,555** training samples in MVSA, we found **143** samples where MNL was inactive (**9.196%**). Among these cases, it is possible that weak-modality advantages existed but were not preserved.
>
> [1]Predictive dynamic fusion. ICML2024

---

> > ### Comment · Reviewer_d8o7 · 2025-08-05
> > **reply**
> >
> > Thanks for providing the detailed reply. most of the questions have been addressed. However after the experiments of MNL (step1) + PDF (step2), the performance gain is still small when comparing with late fusion in table 1. Therefore, I'd like to maintain my score.

---

> > > ### Author Response · Authors · 2025-08-06
> > > **Thank you!**
> > >
> > > Thank you for your feedback and the positive rating. We truly appreciate your valuable insights, which have helped us improve the quality of our work. We will incorporate the suggested experiments and discussions in the revised version. Thank you again for your support of our paper!

---

### Official Review · Reviewer_9fVt · 2025-07-02

**Clarity:** 4
**Significance:** 4
**Originality:** 4
**Rating:** 5
**Confidence:** 5

**Summary:**

This paper introduces a new paradigm for addressing modality imbalance in multimodal learning, called Multimodal Negative Learning (MNL). Unlike traditional positive learning methods that align weak modalities with dominant ones by enhancing target-class prediction, MNL proposes guiding weak modalities to suppress non-target classes, thus stabilizing their decision space. The framework builds on a formal robustness analysis using a newly defined Unimodal Confidence Margin (UCoM), which forms the basis of a provable lower bound on multimodal robustness. MNL is tested under various noise conditions across four datasets and consistently outperforms both static and dynamic late-fusion baselines. The paper is well written, with clear organization, logical flow, and the use of precise and professional terminology throughout.

**Questions:**

* Training complexity analysis.
* Extension to the scenarios with three modalities.

**Ethical Concerns:**

["NO or VERY MINOR ethics concerns only"]

**Final Justification:**

I am satisfied with the authors’ responses, and my concerns have been fully addressed.

**Limitations:**

yes.

**Quality:**

4

**Strengths And Weaknesses:**

**Strength:**
* The concept of **learning not to be** is novel in the multimodal setting and effectively reframes weak modality alignment as negative class suppression rather than target class enhancement. Furthermore, the paper rigorously derives a robustness lower bound for late-fusion systems based on UCoM, providing strong theoretical motivation for the MNL design.
* The proposed Robust Dominant Modality (RDM) definition is dynamic and data-driven, improving generalization and adaptability during training.
* The paper conducts extensive evaluations across multiple datasets and under varying levels of perturbation (e.g., Gaussian, salt noise), with MNL consistently showing performance and robustness gains.
* The paper demonstrates a high standard of academic writing, with coherent organization, a logical progression of ideas, and consistently precise use of domain-specific terminology.

**Weaknesses**:
* The two-stage training scheme (standard CE followed by MNL) adds some complexity to training dynamics and might not be ideal for scenarios requiring fast adaptation.
* It is recommended that the authors provide a more detailed algorithmic workflow or pseudocode to facilitate understanding and reproducibility of the proposed method.
* While MNL is presented for dual-modality cases, its extension to three or more modalities (especially when all are noisy) is not discussed or demonstrated.

---

> ### Author Rebuttal · Authors · 2025-07-31
>
> We sincerely thank you for your valuable comments and appreciate your recognition of our proposed paradigm "Learning not to be" through Multimodal Negative Learning (MNL) and the theoretical foundation based on UCoM. We believe your constructive feedback will help improve the paper and enhance its potential impact on the multimodal learning community.
>
> > W1 & Q1 The two-stage training scheme (standard CE followed by MNL) adds some complexity to training dynamics and might not be ideal for scenarios requiring fast adaptation. & Training complexity analysis.
>
> Thanks for your comment. Integrating MNL with baseline methods introduces minimal computational overhead and does not require any modification to the baseline network architecture. During training, MNL operates only at the output level by identifying the Robust Dominant Modality (RDM) and the Inferior Modality (IM), determining the direction of guidance, and computing the additional loss. During inference, the process remains identical to the baseline and does not introduce any extra cost.
>
> We are glad to report the additional training cost. Specifically, we calculate the average time per training iteration (per batch) on MVSA, FOOD101, NYUDv2, and CREMAD, respectively.
>
>
> | Method                   | MVSA    | FOOD101 | NYUDv2  | CREMAD  |
> | ------------------------ | ------- | ------- | ------- | ------- |
> | STATIC LATE FUSION       | 20.96ms | 69.23ms | 32.27ms | 63.03ms |
> | STATIC LATE FUSION + MNL | 31.15ms | 85.76ms | 49.19ms | 87.11ms |
>
> These results indicate that the additional overhead introduced by MNL is minimal.
>
> > W2 It is recommended that the authors provide a more detailed algorithmic workflow or pseudocode to facilitate understanding and reproducibility of the proposed method.
>
> Thank you for the excellent suggestion. Below, we provide the pseudocode of the algorithm. The inclusion of this pseudocode will be reflected in the final version of the paper.
>
> ```text
> Algorithm 1: Multimodal Learning with MNL
>
> Input:
>  Training set D_train = {(x_i, y_i)}_{i=1}^N ⊂ X × Y
>  Iteration number T, warm-up epoch W
>
> for t = 1 to T do
>     Sample a fresh mini-batch B_t from D_train
>     Feed-forward the batched data B_t to the model
>     Obtain fusion logits and modality-specific logits from the model
>
>     if t > W then
>         Identify RDM and IMs according to Definition 3.3
>         Compute MNL loss according to Eq. (5)
>         Compute total loss according to Eq. (7)
>     else
>         Compute cross-entropy loss
>     end if
>
>     Update model parameters
> end for
> ```
>
> > W3 & Q2 While MNL is presented for dual-modality cases, its extension to three or more modalities (especially when all are noisy) is not discussed or demonstrated.
>
> We conducted additional experiments on the CMU-MOSEI dataset[1], which contains three modalities: text, audio, and visual. In this setting, the Robust Dominant Modality(RDM) is used to guide the other two weaker modalities. The RDM is identified by comparing $P_y$ and $\xi$ (UCoM) across modalities, in accordance with Definition 3.3. This strategy naturally generalizes to scenarios involving more than two modalities.
>
> Table 1 presents the experimental results conducted on the CMU-MOSEI dataset. Gaussian noise for visual modality, blank noise for the text modality, SNR-based noise for audio modality
>
>
> | Method                 | ε = 0 | ε = 5 | ε = 10 |
> | ---------------------- | ------ | ------ | ------- |
> | STATIC LATE FUSION     | 66.42% | 61.71% | 45.80%  |
> | STATIC LATE FUSION+MNL | 67.44% | 63.36% | 52.09%  |
> | PDF                    | 66.14% | 63.54% | 42.47%  |
> | PDF+MNL                | 67.29% | 64.35% | 48.62%  |
>
> Table 2 ablation experiments
>
>
> | prior | confident | robust | ε = 0     | ε = 5     | ε = 10    |
> | ----- | --------- | ------ | ---------- | ---------- | ---------- |
> | √    |           |        | 65.91%     | 60.76%     | 41.01%     |
> |       | √        |        | 67.07%     | 63.87%     | 48.36%     |
> |       | √        | √     | **67.29%** | **64.35%** | **48.62%** |
>
>
> | all-class | non-target | ε = 0     | ε = 5     | ε = 10    |
> | --------- | ---------- | ---------- | ---------- | ---------- |
> | √        |            | 67.24%     | 60.20%     | 39.81%     |
> |           | √         | **67.29%** | **64.35%** | **48.62%** |
>
> Thank you again for this valuable suggestion. We have added the experiment to the manuscript, and it will be included in the final version.
>
> [1]Multibench: Multiscale benchmarks for multimodal representation learning. Advances in neural information processing systems 2021

---

> > ### Comment · Reviewer_9fVt · 2025-08-04
> > **Comments**
> >
> > The authors' response has addressed most of my concerns. The additional experiments demonstrate that integrating the MNL module into the existing method does introduce some computational overhead, even for a single batch. It would be helpful to compare this with similar baseline plug-in modules, if available.

---

> > > ### Author Response · Authors · 2025-08-04
> > > **Further responses to Reviewer 9fVt**
> > >
> > > Thanks for your constructive and timely feedback.
> > >
> > > - We clarify that while MNL adds some computational overhead during training due to the MNL loss, **it incurs no inference overhead since all related components are removed during inference.** Please kindly note that, in our rebuttal, the experiment in response to W1 & Q1 refers to the computational overhead for each batch during the **training phase**, rather than during the **inference phase**.
> > > - Regarding "similar baseline plug-in modules"
> > >
> > >   （1）Gao *et al.* [1] proposed the Stable Unimodal Feature Augmentation (SUFA) module to enhance unimodal robustness. SUFA consists of **a contrastive loss** and a **reparameterization mechanism**.
> > >
> > >   （2）Yang *et al.* [2] proposed CRMT, where CRMT-Step1 introduces **margin regularization** to boost unimodal robustness and can be readily employed as a plug-in module to integrate with the baseline.
> > >
> > > We have added comparisons on the MVSA dataset under the setting of batch size = 16 and Salt noise ($\epsilon=0, 5, 10$). The averaged accuracy and computational overhead per batch **during training phase** are reported in the following Table. Our MNL achieves significantly better performance than competing methods with substantially lower computational overhead. The experimental results suggest the superior effectiveness and efficiency of MNL against the competing plug-in methods.
> > >
> > >
> > > | Method                                      | Train Time (per batch) | Acc@$\epsilon=0$ | Acc@$\epsilon=5$ | Acc@$\epsilon=10$ |
> > > | :------------------------------------------ | :--------------------- | :--------------- | :--------------- | :---------------- |
> > > | STATIC LATE FUSION                          | 20.96ms                | 76.88            | 67.88            | 55.43             |
> > > | STATIC LATE FUSION+SUFA([1] CVPR2024)       | 48.49ms                | 77.11            | 70.05            | 55.93             |
> > > | STATIC LATE FUSION+CRMT-Step1([2] ICLR2024) | 43.37ms                | 78.47            | 74.08            | 61.95             |
> > > | STATIC LATE FUSION+MNL                      | 31.15ms                | 79.50            | 74.68            | 62.31             |
> > >
> > > [1] Embracing Unimodal Aleatoric Uncertainty for Robust Multimodal Fusion. CVPR2024
> > >
> > > [2] Quantifying and enhancing multi-modal robustness with modality preference. ICLR2024

---

> > > > ### Comment · Reviewer_9fVt · 2025-08-05
> > > > **Comments**
> > > >
> > > > I appreciate the authors' timely response and consider my concerns to be adequately addressed.

---

### Official Review · Reviewer_AePS · 2025-07-02

**Clarity:** 3
**Significance:** 3
**Originality:** 3
**Rating:** 4
**Confidence:** 4

**Summary:**

This paper introduces Multimodal Negative Learning (MNL), a framework to address modality imbalance where dominant modalities often overshadow weaker ones. Traditional approaches attempt to force weak modalities to align their predictions with dominant ones, which can suppress the unique information present in the weak modality. In this paper, instead of enhancing the weak modality's ability to predict the correct class, the dominant modality guides it to suppress non-target classes. The authors claim that MNL helps to stabilize the decision space, preserve modality-specific information, and prevent the over-alignment.

The result shows the robustness improvement on multiple benchmarks, especially in noisy and imbalanced conditions.

**Questions:**

Do we have any ablations to the warm-up settings (e.g. steps)? This looks to affect the final model performance much.
What is the additional efficiency cost for MNL training?

minor: the paper’s name looks too general – “Multimodal Negative Learning” sounds to be able to cover the common MLLM tasks, which unfortunately are not verified in this paper

typo in the legend of Figure 1 (b): “vedio” → “video”

**Ethical Concerns:**

["NO or VERY MINOR ethics concerns only"]

**Final Justification:**

My previous major concern was the insufficient evals on common MLLMs. The authors conducted two additional evaluations during the rebuttal process, and the new result looked good to me. Thus I raised my score accordingly.

**Limitations:**

Limitations are not mentioned in this paper.

**Quality:**

3

**Strengths And Weaknesses:**

Strengths

The concept of "Learning Not to be" is novel to the modality imbalance problem. Shifting the learning objective for weak modalities from identifying the right answer to avoiding wrong answers sounds to make sense and insightful.

The paper provides a rigorous theoretical justification.
MNL is tested on four distinct datasets (MVSA, UPMC Food-101, NYU Depth V2, CREMA-D) spanning different task domains and modality combinations. The evaluations are also conducted under multiple noise types and levels.

This paper is well written and easy to read.

Weaknesses

The authors claim modality imbalance is a problem, but there is lack of proof in which conditions such a problem exists and how serious it is. IMO, it’s natural to rely on one modality more than others, for example, self-driving often relies more on visual signals than others (e.g. audio). In MLLM, it’s also common to sometimes rely on one modality but sometimes rely on the other.

This paper also claims MNL can preserve specific information for weak modalities, but there is lack of such proof or analysis as well.

The tasks demonstrated in this paper are all for multimodal classification, which IMO is not commonly used in the real world. Instead, the most common scenario in the ML domain is the MLLMs (and also often “late fusion”, which fits this paper’s scope). It will be more convincing if this paper can show improvement on common MLLM tasks.

Limitations are not mentioned in this paper.

---

> ### Author Rebuttal · Authors · 2025-07-31
>
> We sincerely appreciate the reviewers’ positive recognition of the novelty of our “Learn Not to Be” concept, the soundness of the theoretical justification. Below, we provide detailed responses to your concerns.
>
> > W1  lack of proof in which conditions modality imbalance exists and how serious it is. In MLLM, it’s also common to sometimes rely on one modality but sometimes rely on the other.
>
> Thank you for your valuable comment.
>
> **Multimodal imbalance arises from the model’s limited ability to perceive and utilize weak modalities [1], rather than deficiencies in the input data itself [2].**
>
> - Multimodal imbalance often occurs during model optimization [3], where the learning of one modality suppresses the learning of others.
> - Multimodal imbalance is not related to the unbalanced usage of different modalities, nor does it imply that certain modalities provide less informative data.
> - Even if a modality itself is informative, the model may still fail to make accurate predictions if it cannot effectively exploit the modality’s useful information.
>
> Please kindly note that this notion does not conflict with dynamic fusion strategies that utilize modalities based on data quality.
>
> In our paper, using the CREMA-D dataset as an example, the audio modality alone achieves 60.70% accuracy, and the **visual modality 56.23%**.
>
> - With static late fusion and joint training, audio accuracy drops slightly to 59.35%, while visual accuracy decreases sharply to **41.12%**, demonstrating severe modality imbalance during optimization.
> - However, with MNL guidance, fusion accuracy improves to 73.71%, and the visual modality’s accuracy increases to **58.05%** (>56.23%), surpassing its standalone performance.
>
> > W2 This paper also claims MNL can preserve specific information for weak modalities, but there is lack of such proof or analysis as well.
>
> Thanks for your valuable comments. We have discussed the specific information preservation for weak modalities in Line 298-313 and Figure 4 (a),  Line 51-72 and Figure 2 in Supp.
>
> As shown in **Figure 4(a)**, we compare the average KL divergence between modality predictions and the corresponding performance under different guidance strategies and noise levels.
>
> - All-Class guidance reduces KL divergence by forcing the weaker modality to imitate the stronger one, but this leads to over-alignment and performance degradation.
> - In contrast, MNL applies guidance only on non-target classes, resulting in higher KL divergence and consistently better performance.
>
> MNL achieves improvements of 1.83%, 0.85%, and 4.64% under increasing noise levels. These results indicate that preserving the weaker modality’s distinct information improves robustness and cross-modal complementarity.
>
> In **Supplementary B.1** (shown as pictures),Follow [4] we employ the *dit* toolkit [5] to perform partial information decomposition, aiming to quantify the unique information contributed by each modality. Furthermore, we present a quantitative analysis to support our findings:
>
> Table 1 under Gaussian noise
>
>
> | Method                   | Modality | ε = 0 | ε = 5 | ε = 10 |
> | ------------------------ | -------- | ------ | ------ | ------- |
> | STATIC LATE FUSION       | Audio    | 0.8986 | 0.7848 | 0.4853  |
> |                          | Visual   | 0.0000 | 0.0023 | 0.0063  |
> | STATIC LATE FUSION + MNL | Audio    | 0.4364 | 0.3556 | 0.1221  |
> |                          | Visual   | 0.0076 | 0.0149 | 0.0191  |
>
> Table 2 Under Salt Noise
>
>
> | Method                   | Modality | ε = 0 | ε = 5 | ε = 10 |
> | ------------------------ | -------- | ------ | ------ | ------- |
> | STATIC LATE FUSION       | Audio    | 0.8986 | 0.6150 | 0.4687  |
> |                          | Visual   | 0.0000 | 0.0028 | 0.0037  |
> | STATIC LATE FUSION + MNL | Audio    | 0.4364 | 0.2640 | 0.1073  |
> |                          | Visual   | 0.0076 | 0.0122 | 0.0196  |
>
> As shown in the results, MNL enhances the consistency of weak modality (visual) in expressing its unique information and leads to improved performance. This provides direct evidence and analysis that MNL effectively preserves specific information for weak modalities.
>
> > W3 & Q2 this paper are all for multimodal classification. It will be more convincing if this paper can show improvement on common MLLM tasks. & the paper’s name looks too general
>
> Thank you for your valuable comment.
>
> - Considering that most existing MLLM models [6] primarily perform   feature-level fusion, to demonstrate the generalizability of our approach,**we extend MNL    to large language models**, forming a  multi-LLM fusion setting.
>
>   Specifically, on the MathQA dataset [11], a choice math reasoning  benchmark, we fine-tune InternVL2_5-1B and Qwen2.5-1.5B (which share the same vocabulary) using LoRA. Furthermore, during training, we apply MNL by masking  the correct option and using $P_y$ and UCoM  to allow the stronger model to guide the weaker  one.
>
>   We compare the performance of STATIC LATE FUSION and STATIC LATE FUSION + MNL   to validate the effectiveness of our method. As shown in the table, applying  MNL consistently improves both the individual LLM and the fused result.
>
>
> | Method                   | Fusion | InternVL2_5-1B | Qwen2.5-1.5B |
> | ------------------------ | ------ | -------------- | ------------ |
> | STATIC LATE FUSION       | 50.89  | 42.85          | 49.41        |
> | STATIC LATE FUSION + MNL | 51.42  | 43.32          | 50.85        |
>
> - **MNL belongs to multimodal learning.** Multimodal learning is indeed a sophisticated and widely studied paradigm in the machine learning community and it refers to the integration and joint modeling of data from different modalities such as text, images, speech, video[7]. The aim is to improve a model's understanding capability and generalization performance[8]. This paradigm naturally covers a wide range of tasks, including multimodal classification [9,10], all of which fall under the broad definition of multimodal learning.
>
>   **the core of our method lies in establishing a negative learning paradigm within a multimodal system**. Specifically, the Robust Dominant Modality (RDM) provides supervision on non-target classes to guide the Inferior Modalities (IMs). This strategy not only preserves the unique information inherent to the IMs but also enhances the overall robustness of the multimodal system. Following the reviewers' suggestion, we further validated the effectiveness of MNL in the multiple-LLM setting.
>
>   Therefore, we believe that naming our method **Multimodal Negative Learning** is both appropriate and representative of its core contribution.
>
> > Q1 Do we have any ablations to the warm-up settings (e.g. steps)? This looks to affect the final model performance much. What is the additional efficiency cost for MNL training?
>
> Thank you for your question.
>
> **(1) Warm-up Settings:**
>
> We have provided the ablations on  warm-up epoch. According to the following table, the hyperparameter **warm-up epoch** is not sensitive in practice (we set it to 10). The reason is that the benefit of MNL is relatively limited during the early stages of training. As training progresses and the modality gap becomes more evident, MNL provides more effective guidance. Therefore, the warm-up setting primarily serves to reduce unnecessary computational cost in the early phase of MNL training.
>
>
> | Warm-up Epoch | 0     | 2     | 5     | 10    | 15    | 20    | 30    |
> | ------------- | ----- | ----- | ----- | ----- | ----- | ----- | ----- |
> | Accuracy (%)  | 68.55 | 68.11 | 68.11 | 69.18 | 68.79 | 68.26 | 69.05 |
>
> **(2) Efficiency Cost:**
>
> We provide a detailed efficiency analysis of MNL training’s extra cost by reporting average training iteration time on MVSA, FOOD101, NYUDv2, and CREMA-D, with no added overhead during inference.
>
>
> | Method                   | MVSA    | FOOD101 | NYUDv2  | CREMAD  |
> | ------------------------ | ------- | ------- | ------- | ------- |
> | STATIC LATE FUSION       | 20.96ms | 69.23ms | 32.27ms | 63.03ms |
> | STATIC LATE FUSION + MNL | 31.15ms | 85.76ms | 49.19ms | 87.11ms |
>
> These results indicate that the additional overhead introduced by MNL is minimal.
>
> > Limitations are not mentioned in this paper.
>
> Thanks for your comment.  we have provided a detailed discussion of the limitations Line 123-142 in **Supplementary E**. For example, the additional computational overhead introduced by MNL during training.
>
> > presentation issues
>
> Thank you for pointing this out. We have corrected the typo in the legend of Figure 1(b) and carefully reviewed and revised all representation issues throughout the paper.
>
> And we sincerely hope that our responses have sufficiently addressed your concerns. Should any questions remain, we would be more than glad to provide further clarification. If you find our clarifications satisfactory, we would greatly appreciate your consideration in updating your score to support the acceptance of our paper.
>
> [1]Balanced multimodal learning via on-the-fly gradient modulation. CVPR2022
>
> [2]Missing modality imagination network for emotion recognition with uncertain missing modalities. ACL2021
>
> [3]Balanced Multimodal Learning via On-the-fly Gradient Modulation. CVPR2022
>
> [4]Reconboost:Boosting can achieve modality reconcilement. ICML2024
>
> [5]dit: a Python package for discrete information theory.
>
> [6]Mm-llms: Recent advances in multimodal large language models. ACL2024
>
> [7] Multimodal machine learning: A survey and taxonomy.TPAMI2018
>
> [8]A theory of multimodal learning. NeurIPS2023
>
> [9]Trusted multiview classification. ICLR2021
>
> [10]Multimodal dynamics: Dynamical fusion for trustworthy multimodal classification. CVPR2022
>
> [11]Mathqa: Towards interpretable math word problem solving with operation-based formalisms.

---

> > ### Comment · Reviewer_AePS · 2025-08-05
> >
> > I appreciate the authors' thorough explanation, which has resolved most of my previous concerns.
> >
> > The newly included multi-LLM experiments are promising, and they effectively show the extensibility of MNL to a broader range of multimodal tasks. Nevertheless, the scope of the evaluation remains narrow with only the MathQA task. For a more compelling demonstration of MNL's benefits on widely used MLLMs, if time allows, I would strongly recommend conducting additional experiments on other traditional VLM tasks like captioning, VQA, and document understanding.

---

> > > ### Author Response · Authors · 2025-08-06
> > > **Further Response to Reviewer AePS**
> > >
> > > Thanks for your positive feedback on our experiments. To demonstrate our effectiveness on widely used MLLMs, we have added experiments on the Visual7w dataset [1], which is a **general VQA** dataset and widely used in existing MLLMs [2][3]. In this experiment, due to the computation time, we select 5% data (6993 samples) from the Visual7W dataset as the training set and 1% (1400 samples) for inference. We fine-tune Qwen2-VL-2B-Instruct [4] and Qwen2.5-VL-3B-Instruct [5] with LoRA and compare the performance of STATIC LATE FUSION and STATIC LATE FUSION + MNL to validate the effectiveness of our method. As shown in the table, applying MNL consistently improves both the individual MLLM and the fused results. We will include the complete experiments in the final version. We truly appreciate your valuable comments, which significantly improved the quality of our paper.
> > >
> > >
> > > | Method                   | Fusion | Qwen2-VL-2B-Instruct | Qwen2.5-VL-3B-Instruct |
> > > | ------------------------ | ------ | -------------------- | ---------------------- |
> > > | STATIC LATE FUSION       | 84.87  | 83.71                | 84.76                  |
> > > | STATIC LATE FUSION + MNL | 85.21  | 84.57                | 84.82                  |
> > >
> > > [1] Visual7W: Grounded Question Answering in Images. CVPR 2016
> > >
> > > [2] Cogvlm: Visual expert for pretrained language model. NeurIPS 2024
> > >
> > > [3] Cambrian-1: A fully open, vision-centric exploration of multimodal llms. NeurIPS 2024
> > >
> > > [4] Qwen2-VL: Enhancing Vision-Language Model's Perception of the World at Any Resolution. arXiv 2024
> > >
> > > [5] Qwen2. 5-vl technical report. arXiv 2025

---

> > > > ### Comment · Reviewer_AePS · 2025-08-06
> > > >
> > > > Thank you for the new evaluation, which looks good. As previously suggested, demonstrating MNL's advantages on more common MLLM tasks would greatly increase the paper's impact. I encourage to include these more comprehensive evaluations in the revised version to offer broader insights to the community. I've raised my score accordingly.

---

### Official Review · Reviewer_yEYi · 2025-07-02

**Clarity:** 2
**Significance:** 2
**Originality:** 2
**Rating:** 4
**Confidence:** 3

**Summary:**

The paper addresses the modality imbalance problem, where one modality may dominate others due to noise, limited information, or differences in sensing methods. Rather than forcing weak modalities to align with stronger ones which risks suppressing modality specific information the proposed method aims to preserve the unique information in each modality while fixing the problem created due to modality imbalance.

**Questions:**

-  Robustness radius R(x_i) is greater means the perturbed x_i is far from x_i. You want this small right? If you get larger UCom it means I will have large R(x_i) which is not good right? How does this leads to greater robustness?

- Please refer weakness for more questions.

**Ethical Concerns:**

["NO or VERY MINOR ethics concerns only"]

**Final Justification:**

I appreciate the authors for addressing the concerns regarding clarity and for incorporating the suggested details into the revised version of the paper and response.

I would also encourage the authors to briefly motivate, within the main paper, why late fusion is beneficial compared to latent space-based methods.

Following the above discussion, the authors have agreed to include additional experiments, an introduction to late fusion, a notation table, refined theorem and precise notations for improved clarity in the main paper. If these points are adequately addressed, I believe they will significantly enhance the overall clarity and quality of the work.

Accordingly, I am raising my **clarity and quality scores to 2** each. I would also like to **update my overall score to 4 (Borderline Accept)**, contingent upon the authors incorporating the discussed revisions into the final version.

**Limitations:**

Yes. In conclusion section as a future work.

**Paper Formatting Concerns:**

No formatting issues.

**Quality:**

2

**Strengths And Weaknesses:**

Strengths:
- The idea of preserving the information in weaker modality seems to be novel.
- The paper is driven by the theoretical findings.

Weakness:
- The paper is hard to parse and I have following comments:

      - Abstract (What does it mean to enhance target class to supress non-target class in weak modalities? )

      - Use same word either unimodal or uni-modal (line 25), similarly late fusion or late-fusion.

      - What is late-fusion (line 29) state for readers?

      - Fig 1b) video and guidance spelling wrong?

      - Line 43-50 and Fig 1a) isn't increasing the probability of target class same as decreasing or suppressing the probabilities of non-target class?

      - What does right side of block diagram represent Fig 1a?

      - Fig 1b) Why is video error rate with KL guidance higher than guidance? This means there's no point in using KL guidance at all.

      - Fig 1b) Does audio model also change? The figures are very hard to parse.

- The setting of multimodal learning is not clear at the start. Is the goal to learn a separate classifiers on each modality and effectively combine them or transfer the discriminative power from strong modality to weaker one.
- What are equations in Fig (2) ?
- Theorem and notations are not clear and precise. Properly introduce notations.
- Properly introduce late fusion, it is used everywhere.
- What does decision space instability means?
- What does uncertainty mean? Does it mean lower probability ? Is it same as confidence?
- Hard to parse equation (5)?
- What does $P_y^{(2)}$ means? No definitions provided.
-  What are static and dynamic late fusion? (some intuition would help)
-  Compare with positive learning.
- Results on additional datasets for Table 3 and 4.
- Experimental settings not properly provided with details on hyperparameters.
- What is $\epsilon$ in line 227, Is it gaussian noise variance ?
- If one can segregate the RDM and IM, then why do we need the fusion itself. The RDM is responsible to correct the IM, but why not discard the IM rather than improving it? Probably need an experiment.

---

> ### Author Rebuttal · Authors · 2025-07-31
>
> Thanks for your valuable feedback, which greatly contributed to improving the paper!
>
> > W1: Enhance target class to supress non-target class. & Compare with positive learning.
>
> Thanks for your comment. There maybe some potential misunderstanding.
> **MNL aims to guide the weak modality’s predictions on non-target classes using the dominant modality’s predictions on non-target classes**, rather than constraining the weak modality’s prediction on the target class through the dominant modality’s prediction on the target class. We have added experiments on target class guidance (positive learning) and non-target classes guidance (MNL). As shown in the Table below, the positive learning is not effective enough.
>
> | Method                                | 𝜖 = 0 | 𝜖 = 5 | 𝜖 = 10 |
> |---------------------------------------|--------|--------|---------|
> | STATIC LATE FUSION                    | 76.88  | 63.46  | 55.16   |
> | STATIC LATE FUSION + target class guidance (positive learning) | 77.54  | 63.25  | 55.43   |
> | STATIC LATE FUSION + MNL              | **79.50**  | **74.03**  | **63.01**   |
>
> (1) **Guidance on target class:** Using the dominant modality's prediction on target class to guide that of the weak modality essentially performs single-class guidance, which is similar to applying a soft-label constraint on the target class. This overlaps with the supervision already provided by the cross-entropy loss, thus offering limited additional benefits.
>
> (2) **Guidance on non-target classes:** By leveraging the dominant modality’s predictions on non-target classes to guide those of the weaker modality, we provide the weak modality with richer semantic cues — namely, the dominant modality’s understanding of non-target categories. This effectively enhances the weak modality’s comprehension of non-target classes and improves its overall performance.
>
>
> > W3 & W9 & W12 & W17: Multimodal learning & Late-fusion & Static and dynamic late fusion?
>
> (1) **Multimodal Learning (MML)** refers to learning comprehensive information from multiple modalities and effectively integrating them [1-3], which can be mainly categorized into early, mid-level, and late fusion.
>
> (2) **Late fusion** typically refers to combining the outputs of different models as the final prediction [4-5]. Formally, each modality $x^{(m)}$ is passed through a modality-specific projection function $f^{(m)}$, resulting in an output $z^{(m)} = f^{(m)}(x^{(m)}) \in \mathbb{R}^C$, where $C$ is the number of classes. These outputs are then fused as:
>
> $
> z^{\text{final}} = \sum_{m=1}^M w^{(m)} \cdot z^{(m)}
> $
>
> Here, $w^{(m)}$ is the fusion weight for the $m$-th modality, and $\sum_{m=1}^M w^{(m)} = 1$.
>
> - **Static late fusion** uses fixed weights $w^{(m)}$ across all samples. It assumes the reliability of each modality is consistent.
> - **Dynamic late fusion** allows $w^{(m)}$ to vary across samples, often based on the confidence or uncertainty of each modality’s output. This enables the model to adaptively rely more on the high-quality modality for each input.
>
> > W5: Increasing the probability of target class and suppressing the probabilities of non-target class.
>
> **Using the dominant modality's target class probability to guide that of the weaker modality essentially performs single-class guidance, which is similar to applying a soft-label constraint on the target class.** This overlaps with the supervision already provided by the cross-entropy loss, thus offering limited benefit.
>
> In contrast, **constraining the non-target class predictions of the weaker modality using those from the dominant modality provides richer semantic information**. This complements the label-based cross-entropy loss and effectively enhances the weaker modality’s understanding of non-target classes, thereby improving its overall performance.
>
> It is worth noting that **we do not use the full output distribution of the dominant modality to guide the weaker one, which would risk overriding the unique information captured by the weaker modality**. Instead, we selectively guide only the non-target class predictions, which allows us to inject informative signals from the dominant modality while preserving the individuality of the weaker one.
>
> > W6: Right side of block diagram in Fig 1a.
>
> The weak modality is aligned with the dominant modality across all classes. In contrast, the bottom part of Fig. 1(a) illustrates the dominant modality guiding the weak modality specifically on the non-target classes.
>
> In our formulation, the **specific** information refers to the probability of the target class, while the **common** information refers to the probabilities of the non-target classes.
>
> - **Existing methods (top) tend to align both specific and common information simultaneously**, which may interfere with the unique information of the weak modality.
>
> - **Our method (bottom) constrains only the non-target class predictions**, thereby introducing guidance from the dominant modality while preserving the unique information of the weak modality.
>
> > W7: Video error rate.
>
> The bar chart shows cases where the audio modality errs while video is correct. The line chart tracks how many previously correct video samples become incorrect in the next epoch (Video error rate). This demonstrates that aligning a modality (video) with another across all classes can lead to a  over-alignment collapse.
>
>
> >W8: Does audio model also change?
>
> The audio model changes during training , as both the video and audio models are trained simultaneously.
>
> >W10 & W15: Equations in Fig. 2 & Equation (5).
>
> (1) In **Figure 2**, we illustrate the training losses (CE loss and MNL loss) and highlight MNL **lifts the multimodal robustness lower bound** during optimization.(W10)
>
> (2) Eq. 5 presents the specific form of the MNL loss. It uses the RDM’s predictions to guide IM on non-target classes.
>
> > W13 & W14 : Decision space instability&uncertainty
>
> **Decision space instability** refers to a model's high sensitivity to input perturbations near the decision boundary, where even small changes can lead to significant shifts in prediction. And uncertainty can be intuitively viewed as low confidence.
>
> The weak modality itself is relatively unstable, which, according to Eq. (4), may reduce the overall robustness of the multimodal system after fusion. In contrast, MNL enhances the UCoM of the weak modality, thereby improving the overall robustness of the system.
>
>
> > W19: Results on additional datasets for Table 3 & 4.
>
> We extended our approach and ablation studies to three modalities datasets, the CMU-MOSEI. The results are presented as follow:
>
> | prior | confident | robust | ε = 0  | ε = 5  | ε = 10 |
> |--------|--------|--------|--------|--------|--------|
> | √     |           |        | 65.91% | 60.76% | 41.01% |
> |       | √         |        | 67.07% | 63.87% | 48.36% |
> |       | √         | √      |**67.29%** | **64.35%** | **48.62%** |
>
>
> | all-class | non-target | ε = 0  | ε = 5  | ε = 10 |
> |-----------|------------|--------|--------|--------|
> | √         |            | 67.24% | 60.20% | 39.81% |
> |           | √          |**67.29%** | **64.35%** | **48.62%** |
>
> > W20: Details on hyperparameters.
>
> We have provided some implementation details in Line 215–218 and Line 94-114 of **Supp. C**.
> For the wiegits of CE and MNL losses, the CE weight is fixed at 1 and the MNL weight is varied from 0.2 to 2.0 under different noise levels, we set weight to 1.
>
> |weight    | 0.2    |  0.6    | 1.0        | 1.4    |  2.0    |
> | -------- | ------ |  ------ | ---------- | ------ |  ------ |
> | Gauss_0  | 0.7688 |  0.7919 | **0.8054**   | 0.7900 |  0.7977 |
> | Gauss_5  | 0.7256 |  0.7341 | **0.7407** | 0.7206 |  0.7287 |
> | Gauss_10 | 0.6246 |  0.6266 | **0.6378** | 0.6212 |  0.6370 |
> | Salt_5   | 0.7495 |  0.7437 | **0.7576** | 0.7572 |  0.7526 |
> | Salt_10  | 0.6227 |  0.6346 | **0.6493**     | 0.6408 |  0.6301 |
>
> > W21: $\epsilon$.
>
> We followed the setting of $\epsilon$ in QMF [7] and TMC [8], and unified the noise levels to 0, 5, and 10 for clearer comparison.
>
> For Gaussian noise, it is noise intensity; for salt-and-pepper noise, it is the probability of noise occurrence; for audio, it is the SNR-based noise level; and for text noise, it denotes the probability of blank noise.
>
> > W22: RDM and IM.
>
> - Even if the RDM and IM can be segregated, this does not imply that the performance of the unimodal RDM surpasses that of the fusion of RDM and IM. This is evident from the unimodal versus multimodal fusion results shown in Tables 1 and 2.
>
> - We utilize both RDM and IM to integrate complementary information. Extensive research has shown that multimodal fusion generally outperforms unimodal [6,7].
>
> >Q1: Robustness radius R(x_i).
>
> There might be some misunderstanding here. Our model aims to increase the robustness radius $R(x)$, where a larger UCoM indicates a favorable phenomenon. Specifically, $R(x_i)$ is defined as the *minimum $\ell_2$ perturbation required to flip the model’s prediction from the ground-truth class $y$ to any other class*. A **larger $R(x_i)$** signifies **higher robustness** to input perturbations.
>
> According to Eq. 4, a higher UCoM corresponds to a larger $R(x)$, which means stronger robustness of the multimodal system. MNL leverages the modality with high UCoM to guide the modality with low UCoM, thereby improving the latter’s UCoM and ultimately enhancing the overall system robustness.
>
> > Presentation issues.
>
> We have standardized the use of the terms unimodal and late fusion, corrected the spelling error in Fig. 1(b), and refined Theorem and notations. Finally, we have carefully reviewed and revised all representation issues throughout the paper.
>
> [1] Multimodal machine learning: A survey and taxonomy. TPAMI2018
>
> [2] Multimodal learning with transformers: A survey. TPAMI2023
>
> [3] Multimodal deep learning. ICML2011
>
> [4] PDF. ICML2024
>
> [5] QMF. ICML2023.
>
> [6] DUA-Nets. AAAI2021
>
> [7] CRMT. ICLR2024
>
> [8] OGM-GE. CVPR2022
>
> [9] MMPareto. ICML2024

---

> > ### Comment · Reviewer_yEYi · 2025-08-04
> >
> > I thank the authors for their clarifications and detailed responses to my comments. To further improve the clarity of the paper for readers, I have the following suggestions:
> >
> > - Briefly introduce the concept of late fusion and its different types in the paper.
> > - Introduce the notations using a notation table, and consider including this table in the appendix for easy reference.
> > - Include a comparison table related to positive learning to highlight the necessity of negative learning, not just for MVSA, but also for other datasets.
> >
> > I have one more question:
> >
> > How beneficial is late fusion work like the proposed one compared to learning in the latent space? For instance, one could use a contrastive learning loss or a Deep CCA based loss combined with a task loss (e.g., prediction using shared latent representations or modality specific features). There are existing methods capable of disentangling common and modality specific latent representations, which can then be used for prediction rather than relying on late fusion at the output level.

---

> > > ### Author Response · Authors · 2025-08-05
> > > **Response to Reviewer yEYi (1/3)**
> > >
> > > Thanks for your valuable and timely feedback.
> > >
> > > Due to the character limit in our initial rebuttal, we had to shorten some of our responses, including notations and detailed descriptions. In this revision, we have added further clarifications to  enhance the clarity of the paper for the readers. In addition, we will further address your concerns.
> > >
> > > > The concept of late fusion and its different types
> > >
> > > We appreciate the suggestions and have incorporated the concept of late fusion in the manuscript, which will be included in Section 3 of the final version.
> > >
> > > **The concept of Late Fusion:** In multimodal learning, late fusion is a commonly used fusion strategy that integrates information from different modalities or sensors at the decision level, i.e., after the model's output layer. Unlike early fusion, which combines inputs at the input level, or intermediate fusion, which interacts at the feature level, the core idea of late fusion is to build independent learners for each modality and fuse their outputs as the final prediction.
> > >
> > >  **The detailed description of Late Fusion and its different types** are presented as follows, it will be included in the supplementary material.
> > >
> > > Suppose there are |$\mathcal{M}$| modalities, each associated with a modality-specific learner $f^{(m)}$. Given input modality data $\mathbf{x}^{(m)}$, the $m$-th learner produces a prediction logits vector:
> > >
> > > $$
> > > \mathbf{z}^{(m)} = f^{(m)}(\mathbf{x}^{(m)}) \in \mathbb{R}^C,
> > > $$
> > >
> > > where $C$ denotes the number of classes. Subsequently, the logits vectors from different modalities are fused. The most common fusion strategy is **weighted summation**, defined as:
> > > $$
> > > \mathbf{z} = \sum_{m=1}^{M} w^{(m)} \cdot \mathbf{z}^{(m)}
> > > $$
> > > where $w^{(m)}$ is the weight assigned to modality $m$, $\mathbf{z}^{(m)}$ is the corresponding logits vector and  $\sum_{m=1}^M w^{(m)} = 1$.
> > >
> > > Specifically, **in the case of two modalities**:
> > >
> > > - **Static late fusion** assigns an equal weight of 0.5 to each modality.
> > >
> > > - **Dynamic late fusion** allows $w^{(m)}$ to vary across samples, rather than being fixed.
> > >
> > >
> > >
> > >
> > >
> > >
> > > > Notations using a notation table
> > >
> > > Thank you again for your valuable suggestions. To help readers better understand the paper, we provide a detailed table explaining the symbols.
> > >
> > >
> > > | Symbol               | Description                                                           |
> > > | -------------------- | ------------------------------------------------------------------ |
> > > | $\mathcal{M}$ | Modality Set |
> > > | \|$\mathcal{M}$\| | Number of Modalities |
> > > | $R(x_i)$             | Multimodal robustness radius of the sample $x_i$                    |
> > > | $j$                  | The most probable class among the non-target classes               |
> > > | $f(x_i)$           | Fusion logits output given sample $x_i$                       |
> > > | $f_k(x_i)$           | Fusion logits output for class $k$ given sample $x_i$                       |
> > > | $f^{(m)}(x_i^{(m)})$ | Logits output from the $m$-th modality given modality-specific input |
> > > | $\sigma(\cdot)$      | Softmax function                                                   |
> > > | $\xi_{(m)}^j$          |UCoM for modality $m$ as defined in Eq. (2) and j denotes the most probable class among the non-target classes. |
> > > | $\xi_{(m)}$          | UCoM for modality $m$, in its simplified form, is denoted by $\xi_{(m)}^j$ without explicitly specifying the competing class $j$.  |
> > > | $\xi_{(m)}'$         | UCoM for modality $m$ after perturbation                                            |
> > > | $\tau_{(m)}$         | Lipschitz constant of modality $m$                                  |
> > > | $w^{(m)}$            | Weight assigned to modality $m$ in late fusion                      |
> > > | $P^{(m)}$              | The predicted probability of modality $m$      |
> > > | $P_y^{(m)}$              | The predicted probability of the target class from modality $m$      |

---

> > > > ### Author Response · Authors · 2025-08-05
> > > > **Response to Reviewer yEYi (2/3)**
> > > >
> > > > > A comparison table related to positive learning on three additional datasets.
> > > >
> > > > To highlight the necessity of negative learning, we have added experiments on the UMPC FOOD 101, NYU DEPTHV2, and CREMA-D datasets under varying levels of Gaussian noise, as shown in Tables 1–3. The results clearly demonstrate that providing only target class guidance yields minimal improvement. This underscores the superior performance and importance of our proposed multimodal negative learning (MNL), which is theoretically supported by its ability to improve the multimodal robustness lower bound. All experiments will be included  in the final version.
> > > >
> > > > Tabel 1 Comparisons on UMPC FOOD 101 under Gaussian noise
> > > > | Method                                             | $\epsilon=0$ | $\epsilon=5$ | $\epsilon=10$ |
> > > > |----------------------------------------------------|--------------|--------------|---------------|
> > > > | STATIC LATE FUSION                                 | 90.69        | 68.49        | 57.99         |
> > > > | STATIC LATE FUSION + target class guidance (positive learning) | 91.82        | 69.17        | 56.51         |
> > > > | STATIC LATE FUSION + MNL                           | **92.77**        | **75.16**        | **62.06**        |
> > > >
> > > > Tabel 2 Comparisons on NYU DEPTHV2 under Gaussian noise
> > > > | Method                                             | $\epsilon=0$ | $\epsilon=5$ | $\epsilon=10$ |
> > > > |----------------------------------------------------|--------------|--------------|---------------|
> > > > | STATIC LATE FUSION                                 | 70.03        | 64.37        | 60.55         |
> > > > | STATIC LATE FUSION + target class guidance (positive learning) | 70.79        | 63.61        | 61.33         |
> > > > | STATIC LATE FUSION + MNL                           | **71.05**        | **67.02**        | **63.81**         |
> > > >
> > > > Tabel 3 Comparisons on CREMA-D under Gaussian noise
> > > > | Method                                             | $\epsilon=0$ | $\epsilon=5$ | $\epsilon=10$ |
> > > > |----------------------------------------------------|--------------|--------------|---------------|
> > > > | STATIC LATE FUSION                                 | 68.04        | 64.25        | 52.39         |
> > > > | STATIC LATE FUSION + target class guidance (positive learning) | 69.96        | 64.78        | 55.64         |
> > > > | STATIC LATE FUSION + MNL                           | **73.71**        | **70.35**        | **57.26**         |

---

> > > > > ### Author Response · Authors · 2025-08-05
> > > > > **Response to Reviewer yEYi (3/3)**
> > > > >
> > > > > > How beneficial is late fusion work like the proposed one compared to learning in the latent space?  For instance, one could use a contrastive learning loss or a Deep CCA based loss combined with a task loss (e.g., prediction using shared latent representations or modality specific features). There are existing methods capable of disentangling common and modality specific latent representations, which can then be used for prediction rather than relying on late fusion at the output level.
> > > > >
> > > > >
> > > > >
> > > > > (1) In multimodal learning, the advantages of late fusion lie in its flexibility, robustness, and interpretability [1].
> > > > > - **Flexibility**. Latent fusion requires aligning and projecting heterogeneous modalities into a unified representation space, which is often challenging and prone to noise or information loss. In contrast, late fusion processes each modality independently and combines only the final predictions, significantly reducing the need for strict alignment. As a result, late fusion is more flexible and better suited for integrating diverse multimodal data types.
> > > > >
> > > > > - **Robustness**. In real-world scenarios, data from one modality may be missing entirely or severely degraded (e.g., due to noise). In latent fusion, such incomplete or low-quality modalities can corrupt the shared representation, leading to a significant drop in performance [1][7]. In contrast, late fusion treats each modality independently and fuses only the final predictions. As a result, if one modality is missing, its output can be simply ignored, and if one modality is noisy, it does not directly affect the others. This modular structure makes late fusion inherently more robust to missing or noisy modalities.
> > > > >
> > > > > - **Interpretability**. Latent fusion occurs at the intermediate stages of the model, where fused features are often highly abstract, making it difficult to clearly understand the specific contribution of each modality to the final decision. In contrast, late fusion combines individual modality predictions, allowing inspection of each modality’s output separately. This enables easier identification of which modality’s prediction caused an error, thereby offering superior interpretability.
> > > > >
> > > > >
> > > > > (2) We have further added comparisons with latent fusion methods on the MVSA dataset. The experiments are conducted under varying levels of Gaussian noise, and all the competing methods share the same backbone. We detailed these methods as follows:
> > > > >
> > > > > - Deep CCA: We followed [3][4] to implement the CCA loss and combined it with a cross-entropy loss, which performs using the shared latent representation.
> > > > > - LFM [5]: LFM integrates  **contrastive learning** into multimodal learning to align features across modalities.
> > > > > - SUFA [6]: SUFA aligns different modalities by **minimizing the KL divergence** between them. In addition, it constructs positive and negative sample pairs within each unimodal branch for **contrastive learning**.
> > > > >
> > > > >
> > > > >  The results consistently show that MNL achieves superior performance, especially under noisy conditions, which validates our effectiveness and robustness.
> > > > >
> > > > >
> > > > > Table 4 Compared with latent fusion
> > > > > | Method                      | ϵ = 0 | ϵ = 5 | ϵ = 10 |
> > > > > |----------------------------|-------|-------|--------|
> > > > > | Deep CCA                   | 76.49 | 63.58 | 53.94  |
> > > > > | LFM ([5] NeurIPS2024)                        | 77.26 | 66.47 | 57.80  |
> > > > > | SUFA ([6] CVPR2024)                       | 78.18 | 71.40 | 59.11  |
> > > > > |  MNL   | 79.50 | 74.03 | 63.01  |
> > > > >
> > > > >
> > > > >
> > > > > [1] Multimodal machine learning: A survey and taxonomy. TPAMI2018
> > > > >
> > > > > [2] Multi-View 3D Object Detection Network for Autonomous Driving. CVPR 2017
> > > > >
> > > > > [3] Deep Canonical Correlation Analysis. ICML2013
> > > > >
> > > > > [4] On Deep Multi-View Representation Learning. ICML2015
> > > > >
> > > > > [5] Facilitating Multimodal Classification via Dynamically Learning Modality Gap. NeurIPS2024
> > > > >
> > > > > [6]  Embracing Unimodal Aleatoric Uncertainty for Robust Multimodal Fusion. CVPR2024
> > > > >
> > > > > [7] Are Multimodal Transformers Robust to Missing Modality? CVOR2022

---

> > > > > > ### Comment · Reviewer_yEYi · 2025-08-05
> > > > > >
> > > > > > I appreciate the authors for addressing the concerns regarding clarity and for incorporating the suggested details into the revised version of the paper and response.
> > > > > >
> > > > > > I would also encourage the authors to briefly motivate, within the main paper, why late fusion is beneficial compared to latent space-based methods.
> > > > > >
> > > > > > Following the above discussion, the authors have agreed to include additional experiments, an introduction to late fusion, a notation table, refined theorem and precise notations for improved clarity in the main paper. If these points are adequately addressed, I believe they will significantly enhance the overall clarity and quality of the work.
> > > > > >
> > > > > > Accordingly, I am raising my **clarity and quality scores to 2** each. I would also like to **update my overall score to 4 (Borderline Accept)**, contingent upon the authors incorporating the discussed revisions into the final version.

---

> > > > > > > ### Author Response · Authors · 2025-08-07
> > > > > > > **Thank you!**
> > > > > > >
> > > > > > > Thanks for your positive feedback and raising the score.
> > > > > > >
> > > > > > > We truly appreciate your constructive suggestions and open discussion. We will incorporate all the discussed revisions to further improve the clarity and quality of the main paper. If you have any further questions or suggestions, please feel free to post, and we will continually work on this project and actively address your concerns.
> > > > > > >
> > > > > > > Best regards,
> > > > > > >
> > > > > > > Authors of Submission 10186

---

### Note · Authors · 2025-08-15

Dear SACs, ACs, and Reviewers,

We sincerely thank the SACs, ACs, and reviewers for the valuable feedback and insightful reviews, which have strengthened our paper. The discussions have been highly productive, leading to clarifications and experiments that addressed reviewers' concerns.

We propose a novel multimodal negative learning (MNL) framework, grounded in theoretical guarantees and centered on the perspective of “Learning not to be”. We are glad that the reviewers found our contributions impressive. Our contributions are listed as follows:

- We provide a new insight into multimodal late fusion from the perspective of robustness.

- We introduce a multimodal negative learning method with theoretical guarantees to dynamically enhance the capability of weak modality.

- We conduct extensive experiments to demonstrate the effectiveness of our method without additional inference cost.

While multiple reviewers recognized the novelty and effectiveness of our approach, we have included expanded experiments and detailed explanations in response to their valuable feedback to improve clarity and understanding.

Expanded Experiments: We extended our experiments to a three-modality setting, applied MNL to LLMs (QA) and MLLMs (VQA), analyzed its advantages over latent fusion methods and its training-time computational overhead.

Detailed Explanations: We provide more definitions of multimodal learning, multimodal imbalance, and MNL, along with detailed hyperparameter settings, a notation table, and algorithm pseudocode to aid understanding.

Finally, thanks again for your time and expertise. We understand the constraints of time and workload that reviewers and AC face, and we appreciate the  effort already put into evaluating our work. We will enhance the final manuscript by incorporating these additional insights and analyses.

Best regards,

Authors of submission 10186.

---

### Decision · Program_Chairs · 2025-09-17

**Decision:**

Accept (poster)

**Comment:**

**Paper Summary:**\
This paper introduces a novel multimodal learning paradigm for addressing modality imbalance in multimodal learning. The concept of 'learning not to be' introduced in the paper is novel, and the experiments provide strong support for the authors’ claims. The paper received five peer review comments, and the authors engaged in several rounds of in-depth discussions with the reviewers. Based on these comments, most issues have been addressed, and the paper received unanimous recommendations for acceptance.

**Justification:**\
Given that the reviewers engaged in in-depth discussions with the authors and unanimously recommended acceptance, I concur with their assessment.

**Summary of Rebuttal Period:**\
Reviewers yEYi, 9fVt, and v4mL offered suggestions regarding the paper’s narrative style. Reviewer AePS raised questions about the experimental setup, while Reviewers AePS and 9fVt expressed concerns about scalability. Reviewer 9fVt also noted issues with complexity, and Reviewers d8o7 and v4mL raised questions about the theoretical foundation. Overall, the paper has been thoroughly discussed, and the reviewers’ concerns have been satisfactorily addressed.